

# Hydrological threats for riparian wetlands of international importance – a global quantitative and qualitative analysis

Christof Schneider[1], Martina Flörke[1], Lucia De Stefano[2], Jacob D. Petersen-Perlman[3]

[1] Center for Environmental Systems Research, University of Kassel, Kassel, Germany

[2] Department of Geodynamics, Complutense University of Madrid, Madrid, Spain

[3] Department of Geography, Western Oregon University, Monmouth, Oregon, USA

*Correspondence to*: Christof Schneider (schneider@usf.uni-kassel.de)

**Abstract.** Riparian wetlands have been reportedly disappearing at an accelerating rate. Their ecological integrity as well as their vital ecosystem services for mankind depend on regular inundation patterns of natural flow regimes. However, river hydrology has been altered worldwide. Dams cause less variable flow regimes and water abstractions decrease the amount of flow so that ecologically important flood pulses are often reduced. Given growing population pressure and projected climate change, immediate action is required. Adaptive dam management, sophisticated environmental flow provisions, water use efficiency enhancement, and improved flood management plans are necessary for a sustainable path into the future. Their implementation, however, is often a complex task. This paper aims at identifying hydrological threats for 93 Ramsar sites, many of which are located in transboundary basins. First, the WaterGAP3 modeling framework is used to quantitatively compare current and future modified flow regimes to natural flow conditions. Results show that current water resource management seriously impairs riparian wetland inundation at 29% of the analyzed sites. Further 8% experience significantly reduced flood pulses. In the future, Eastern Europe, Western Asia as well as central South America could be hotspots of further flow modifications due to climate change. Second, impacts on riparian wetland flooding are qualitatively assessed. New dam initiatives in the upstream areas were compiled to estimate the potential for future flow modifications. They currently take place in one third of the upstream areas and are likely to impair especially wetlands located in South America, Africa, Asia and the Balkan Peninsula. Further qualitative results address the capacity to act for each site by evaluating whether upstream water resource availability and the existing legal and institutional framework could support the implementation of conservation measures.

## 1     Introduction

On a global scale, 64-71% of all wetlands have been lost since 1900 (Davidson, 2014) and even higher numbers are expected for floodplain wetlands. For example, in Europe and North America up to 90% of all floodplains are functionally extinct and in developing countries they are disappearing at an accelerating rate (Tockner and Stanford, 2002). Moreover, river systems



belong to the most threatened ecosystems on the planet and the global freshwater Living Planet Index, indicating changes of fish, bird, reptile, amphibian and mammal populations, declined by 76% since 1970 (WWF, 2014). One of the main reasons for this situation is the alteration of natural flow regimes (including natural inundation patterns) due to water resource development (Dynesius and Nilsson, 1994; Kingsford, 2000; Tockner and Stanford, 2002). Dams are built for different purposes such as water supply, hydropower generation, flood control, and navigation. On the one hand, dams offer important benefits and contribute to 12-16% of global food production and 19% of global electricity generation (WCD, 2000; Richter and Thomas, 2007). On the other hand, dams have been identified as the largest anthropogenic impact on the natural environment (Petts, 1984; Dynesius and Nilsson, 1994; Poff et al., 1997). A study of Nilsson et al. (2005) showed that on the global scale dams affect 59% of all large (i.e. virgin mean annual discharge $\geq 350 m^3/s$) river systems. In the year 2000, the total cumulative storage capacity of large dams accounted for approximately 8300 $km^3$ (Chao et al., 2008; ICOLD, 2007), so that more than 20% of global annual river discharge can be retained in reservoirs (Vörösmarty et al., 1997). In general, dams cause less variable flow regimes by considerably dampening flood peaks and elevating low flows. The downstream effects of individual dams are felt for tens to hundreds of kilometers, reducing the extent and frequency of floodplain wetland inundation (Collier et al., 1996; McCully, 1996; Poff et al., 2007). Further decreases in flow are caused by water abstractions of an exponentially growing world population. In the year 2003, 3856 $km^3$ of freshwater were withdrawn globally according to AQUASTAT statistics (FAO, 2010). The main fraction was used by agriculture (70%) followed by industrial (19%) and domestic water supply sectors (11%). Often, man-made infrastructure is required to transfer water and fulfil water demands of different sectors. In particular large cities, which spatially concentrate freshwater demands of millions of people into small areas, currently divert 184 $km^3$ of water over a cumulative distance of 27000 km (McDonald et al., 2014) causing flow alterations through inter- and intra-basin transfers. River flow regime modifications by dams, abstractions and diversions have come at great costs (WCD, 2000; WWF, 2004; Richter and Thomas, 2007). Reviewing 165 case studies, Poff and Zimmermann (2010) demonstrated that alterations of natural flows lead to ecological consequences. In 92% of the cases, ecological impacts were reported. Similar outcomes (86% of 65 case studies) were found by Lloyd et al. (2004).

While floods are known as one of the most damaging natural disasters worldwide affecting human lives and property (Jonkman, 2005; Doocy et al., 2013; Swiss Re 2014), they are also essential at natural sites and benefit river-floodplain ecosystems and their socio-economic functions for society. In this paper we emphasize that a new approach is needed to water resource management. This approach should include not only flood protection for people but also the allowance of sufficient high flows for sustaining floodplain wetlands which also provide vital services to society.

A natural river floodplain represents an ecotone at the interface of aquatic and terrestrial realms, which is periodically flooded and dried, and falls into the wetland category (Gregory et al., 1991; Bayley, 1995). Here, flows above bankfull are by far the single most important driving force (Welcomme, 1979; Junk et al., 1989; Tockner and Stanford, 2002) initiating ecological processes, shaping habitat structures, and causing an important exchange of water, organisms, organic matter and



inorganic nutrients (Matthews and Richter, 2007). All characteristics and interactions caused by flooding are described by the flood pulse concept (Junk et al., 1989; Bayley, 1991; Tockner et al., 2000; Junk and Wantzen, 2004) and engender one of the most dynamic, diverse and productive systems in the world (Naiman et al., 1993; Nilsson and Berggren, 2000). Not only do floodplain wetlands contain more species than any other landscape unit (Tockner and Stanford, 2002; Allan et al., 2005),

they also possess a disproportionately high number of rare species and community types (Nislow et al., 2002). As hotspots of biodiversity, they provide vital ecosystem services for society and economy: (i) The high productivity of floodplains generates important resources such as wood, reed, hay and fish. Their soils are very fertile due to the regular enrichment of nutrient-rich sediments, and floodplain fishery is an important source of protein and income for millions of people, especially in tropical countries (Bayley, 1991). (ii) Decomposition rates of floodplain wetlands are high as well. They act as

biogeochemical reactors that improve water quality during inundation by removing nutrients and toxins. Thus, they help to buffer non-point-source pollution (Dynesius and Nilsson, 1994). (iii) Aesthetic and recreational values support human leisure time activities such as fishing, hunting, hiking and wildlife watching. (iv) The rich genetic and species diversity ensures ecosystem integrity and serves as raw material for adaptation, evolution and medical research, and (v) access to and inundation of natural floodplains buffer extreme hydrological events and hence, help to avoid flood damages. Costanza et al.

(1997) estimated the monetary value of ecosystem services from floodplains and swamps worldwide at US$ 3231 billion per year, and more recently, a global economic assessment of 'The Economics of Ecosystems & Biodiversity' (TEEB) determined the value of the world's wetlands at US$3.4 billion per year (Brander and Schuyt, 2010). Despite the adherent uncertainty of these numbers, they show that next to ecological, social and cultural benefits, also financial gains can be achieved by floodplain conservation methods.

Due to population growth, climate change and new dam initiatives, impacts on riparian wetlands are very likely to further increase in the next decades. Currently, major initiatives in hydropower development are taking place as a new source of renewable energy. At least 3700 major dams are either planned or under construction, which is supposed to reduce the number of remaining free-flowing rivers by further 21% (Zarfl et al., 2014). These dams offer economic opportunities, but have the potential to negatively impact river ecosystem health and cause conflicts among fellow riparians. Climate change

may severely alter flow regimes over large regional scales as well (Nohara et al., 2006; Laize et al., 2014). Projections indicate that future flow regimes are likely to be different due to regionally and seasonally changing precipitation patterns and amounts. Additionally, the higher temperatures influence timing and quantities of snowmelt (Verzano and Menzel, 2009), frequency and intensity of extreme weather events such as floods (Milly et al., 2008), as well as transpiration by plants and evaporation from surfaces (Frederick and Major, 1997; IPCC, 2007). Depending on the applied scenario,

Okruszko et al. (2011) showed that European wetlands could lose 26 to 46% of their ecosystem services by 2050 due to climatic and socioeconomic impacts on hydrology. Today, a strong consensus exists among scientists that (i) natural flow variability needs to be maintained to some degree to preserve river ecosystems and the goods and services they provide (Poff et al., 1997; Postel and Richter, 2003; Arthington et al., 2006; Richter, 2009), and (ii) ecosystems should be considered as



'legitimate users', whose water requirements should be taken into account in allocation schemes in-line with other water use sectors (Naiman et al., 2002; Postel and Richter, 2003; Poff and Matthews, 2013). Measures encompass adaptive integrated dam management that reconciles interests of different water use sectors, water use-efficiency enhancement, and sophisticated environmental flow (eFlow) provisions, e.g. according to the Block Building Methodology (BBM; Tharme and King, 1998)

or the Basic Flow Methodology (BFM; Palau and Alcazar, 2012). These methodologies respect ecologically relevant flow elements such as flood pulses for riparian wetlands. Additionally, questions are being asked about cost and effectiveness of current flood and floodplain management policies, and the potential of restoring river floodplains and dead stream branches to minimize flood damages (Sparks, 1995). However, implementing such measures is a complex task and faces challenges such as setting strategic goals, identifying operation targets, having conflict resolution mechanisms in place, involving

stakeholders, and monitoring the entire development (Pahl-Wostl et al., 2013). International reviews (Moore, 2004; Le Quesne et al., 2010) revealed that the main obstacles for environmental flow implementations around the world include insufficient legal and institutional capacities, as well as conflicts of interests regarding available water resources. This is especially the case in transboundary river basins. The more countries affect the water management upstream of a riparian wetland, the more groups of stakeholders with different interests are present, the higher the potential for conflicts, and the

more interdependencies are created at different administrative levels both within and between the countries (GWP, 2014). Hence, international water treaties and institutions are required to agree on common goals, coordinate basin-wide water management and allocate water to different users (Le Quesne et al., 2010). In the past, ineffective governance systems have often led to overexploitation of water resources with detrimental effects for river ecosystems and, in the long-term, for human well-being (Pahl-Wostl et al., 2013).

Despite the political and legal progress in recent years, water provisions for river ecosystems are still assigned a low priority in water management (Poff et al., 1997; Revenga et al. 2000; Smakhtin et al., 2004), much less funds have been invested into river ecosystem conservation in comparison to human water security (GEF, 2008; Vörösmarty et al., 2010), and in many countries ecological water requirements have not been assessed yet (Smakhtin and Eriyagama, 2008; Richter, 2009). Thus, most river reaches and wetlands worldwide remain vulnerable to overexploitation (Poff et al., 2009; Richter et al., 2011).

Case studies show that floodplain wetlands have been downsized and transformed into terrestrial ecosystems due to reduced flooding caused by water resource management (Hughes, 1988; Maheshwari et al., 1995; Barbier and Thompson, 1998; Kingsford, 2000; Nislow et al., 2002).

In this context, the goal of this study is to identify riparian wetlands that are threatened due to modification of inundation regimes. Thereby, the following research questions are addressed:

1. What is the impact of current water resource management on riparian wetland flooding? Thereby this study considers operation (rather than reservoir capacity and river fragmentation) of 6025 large dams, distinguishes



operation schemes of different dam types, and takes into account water consumption of five different water use sectors including water transfers of larger cities.

2. At which sites is inundation likely to be further impaired in the future? Therefore, this study quantifies the impact of climate change on future flood pulses and compiles major dam initiatives upstream of each wetland.

3. At which sites the implementation of conservation measures could be hindered by a low capacity to act? Therefore, upstream water resource availability as well as the presence of institutional arrangements to facilitate the establishment of eFlows are assessed.

## 2        Methodology

This study focuses on the analysis of a selected sample of riparian wetlands of international importance which are listed under the Ramsar Convention, a global framework for intergovernmental cooperation aiming for the conservation and sustainable use of wetlands. The Ramsar Classification System describes different wetland types, but does not categorize floodplain wetlands as a specific wetland type. Hence, Ramsar wetlands were taken into account, which mainly depend on lateral overspill of adjacent rivers (i.e. fluviogenic wetlands) according to information provided by the Ramsar information sheets (RSIS, no date). For Europe, a higher number of sites were gained as the European wetland geodatabase (Okruszko et al., 2011) clearly defines wetland type and main source of water for each European Ramsar wetland. Altogether, 93 sites were selected ranging from 5 to 55374 km$^2$ in size. They are located in 48 different countries and 47 different river basins. The highest number was found in the Danube basin, with 19 riparian wetlands of international importance. A detailed list of all wetlands is provided in the Supplement.

Today, riparian wetlands are at risk due to dam and water management practices that make river flows less variable and reduce lateral overspill to the adjacent floodplains. Further impairments on the river flow can be expected by climate change. Hence, in a first step, we conducted a quantitative analysis based on the flood pulse concept which describes the flood pulse as a major driver determining the extent of the river floodplain and the biota living within it (Junk et al., 1989; Tockner et al., 2000). For each site we determined the percentage change in flood volume due to (i) current dam operation and water consumption of five different water use sectors including water transfers of larger cities and (ii) climate change projections for the 2050s. Thereby, we compared the modified river flow regimes to natural flow conditions. The natural flow was simulated taking into account current climate and land-cover conditions, but no further anthropogenic impacts. In a second step, a qualitative assessment was conducted addressing threats by new dam initiatives and missing capacity to act. New dam initiatives have the potential to further reduce wetland inundation in the near future. Sufficient capacity to act is necessary to implement counteractive measures at threatened sites and equitably allocated water resources to the different water use sectors.



## 2.1  The quantitative assessment of threats

In order to quantitatively assess anthropogenic alterations of flood pulses at Ramsar sites, the following procedure was taken: (i) simulation of daily natural flow regimes for the time period 1981-2010, (ii) simulation of daily flow regimes for the same time period modified by current water resource management, (iii) simulation of daily flow regimes for the 2050s according to climate change projections, (iv) estimation of bankfull flow as a crucial threshold that marks the starting point of inundation, (v) analysis of all overbank flows by calculating the mean annual flood volume for the modified and the natural flow regimes, and (vi) determining the deviations in flood volume in the modified flow regimes in comparison to natural flow conditions.

For the simulation of daily river discharge, the integrated WaterGAP3 modeling framework was applied (Verzano, 2009), which performs its calculations on a global 5 x 5 arc minute grid cell raster (~9 x 9 km$^2$ at the Equator). The global hydrology model of WaterGAP3 computes the macro-scale behavior of the terrestrial water cycle. In order to run it for the time period 1981-2010, the WATCH-Forcing-Data-ERA-Interim (WFDEI; Weedon et al., 2014) were used as climate input. It consists of a set of daily, 0.5 x 0.5 degree gridded meteorological forcing data, which were simply disaggregated to the 5 arc minute resolution as required by the model. Forced by the climatic time series, WaterGAP3 calculates daily water balances for each grid cell taking into account distributed physiographic characteristics from high spatial resolution maps describing slope, soil type, land cover, aquifer type, permafrost and glaciers, as well as extent and location of lakes and wetlands. The total runoff in each grid cell, derived from the water balances of land and freshwater areas, is routed along a predefined drainage direction map (DDM5; Lehner et al., 2008) to the catchment outlet. Land cover data were derived from the Global Land Cover Characterization map (GLCC; USGS, 2008) and for EU countries from the CORINE Land Cover map (CLC2000; EEA, 2004). This entire setting was used to gain near-natural flow regimes.

For the flow regimes modified by current water resource management, additionally, anthropogenic flow alterations due to water use and dam operation were taken into account. For these model runs, the natural discharge was reduced in each grid cell by consumptive water use as calculated by the global water use models of WaterGAP3. These models simulate spatially distributed sectoral water uses for the five most important water use sectors: electricity production, manufacturing, domestic use, agricultural crop irrigation and livestock watering (Aus der Beek et al., 2010; Flörke et al., 2013). Assuming an optimal water supply to irrigated crops, net irrigation requirements are simulated for each grid cell based on climatic conditions, dominant crop type and irrigated area around the year 2005 (GMIAv5; Siebert et al., 2013). Livestock water demands are determined by multiplying the number of animals per grid cell by the livestock-specific water use intensity (Alcamo et al., 2003). The amount of cooling water consumed by the electricity production sector is calculated by multiplying the water use intensity of each power station with the equivalent annual thermal electricity production. The water use intensity is affected by the cooling system (once-through flow cooling, tower cooling, or ponds) and the type of fuel (coal and petroleum, natural gas and oil, nuclear, or biomass and waste) used at each power station (Flörke et al., 2012). Power station characteristics



such as type, size and location are derived from the World Electric Power Plants Data Set (UDI, 2004). Consumptive water uses of the manufacturing and domestic sectors are computed on a country scale following data from national statistics and reports, which are subsequently allocated to the grid cells of the associated country by means of urban population and population density maps, respectively (Flörke et al., 2013). For the domestic sector, WaterGAP3 also considers the water

transfers of 480 larger cities including their 1642 withdrawal points (City Water Map; McDonald et al., 2014).

In order to assess flow alterations due to dam operation, 6025 large dams with a total storage capacity of 6200 km$^3$ were allocated to the WaterGAP3 stream net based on information of the Global Reservoir and Dam (GRanD) database (Lehner et al., 2011). The criteria of implementation were a minimum dam height of 15 meters which is in accordance with the ICOLD definition for large dams. Additionally, dams exceeding a reservoir storage volume of 0.5 km³ were considered even with a

lower dam height. The operation of dams is performed in WaterGAP3 as a function of dam type. All dams with the main purpose for irrigation are operated according to the algorithm of Hanasaki et al. (2006) with minor modifications by Döll et al. (2009). In the algorithm, annual reservoir release is a function of long-term annual reservoir inflow, the water balance over the reservoir, and the relative reservoir storage at the beginning of the operational year. Subsequently, monthly reservoir releases are calculated depending on the downstream consumptive water use in each month.

All other dam types are operated now in WaterGAP3 based on an optimization scheme provided by Van Beek et al. (2011). Depending on the dam type, an objective function is applied that maximizes electricity production by maximizing the hydrostatic pressure head to the turbines (hydropower dams), minimizes flood damages by minimizing overbank flows (flood control dams), and aims for a constant outflow by minimising deviations from the annual mean (water supply and navigation dams). Furthermore, different constraints are considered which reserve sufficient storage capacity to

accommodate larger floods for seven days (flood protection) and keep sufficient water in the reservoir to safeguard a minimum flow for at least thirty days (minimum flow provisions). Given current reservoir storage and monthly inflow data of the upcoming year, the overall modelling strategy is to find the monthly target storages (and corresponding monthly reservoir releases) that would ensure optimal functioning of the dam. This strategy was realized in WaterGAP3 by evaluating objective functions and constraints through deterministic dynamic optimization (Bellman, 1957) and discretizing reservoir

storage by the Savarenskiy's scheme (Savarenskiy, 1940) considering a discretization width of 2%. At the beginning of each month, the accumulated objective function value is computed for the upcoming twelve months taking into account every possible combination of the discrete reservoir storage classes. The combination, which provides the most suitable value for the objective function without harming any constraint, determines the dam operation scheme. As inflow data, forecasted monthly values are used derived from average simulated flows of the last 5 years (rather than simulated values for the future

year). This prospective scheme reflects more realistically the hydrological situation, where water managers have to deal with uncertain forecast as well (van Beek et al., 2011). The monthly target storages together with the actual incoming flow are subsequently used to calculate the daily reservoir releases. At about 1600 gauging stations globally, the simulated river flow





is calibrated against observed annual river flow data from the Global Runoff Data Centre (GRDC, 2004). The calibration process adjusts only one model parameter, which has an effect on cell surface runoff generation at gauging stations (Döll et al., 2003).

For the flow regimes modified by climate change, additional WaterGAP3 runs were conducted with bias-corrected, daily
climate data from five different general circulation models (GFDL-ESM2M, HadGEM2-ES, IPSL-CM5A-LR, MIROC-ESM-CHEM, and NorESM1-M) taken from ISI-MIP (Hempel et al., 2013). Thereby, we assumed climate drivers to follow the Representative Concentration Pathway leading to a radiative forcing (cumulative measure of human emissions of GHGs from all sources) of 6.0 W/m² (RCP6.0), which depicts a medium-high emission scenario with stabilization from 2050 onwards (Riahi et al., 2011). River discharge was modelled for the 2050s (represented by the time period 2041-2070) but
also for the baseline period 1981-2010 to gain GCM-driven natural flow regimes as reference condition. In order to focus on the exclusive effect of climate change, dam operation and water use were disabled in these model runs.

An important parameter in our analysis is bankfull flow as it describes the point where the flow starts to enter the active floodplain. Bankfull flow was estimated for each single grid cell by the partial duration series approach taking into account 30-year time series of daily discharge data modelled by WaterGAP3 and applying an increasing threshold censoring
procedure, a declustering scheme and the generalized Pareto distribution. The approach including a validation of bankfull flow estimates is described in detail by Schneider et al. (2011a). As floodplain inundation requires overtopping of the banks, each daily flow above bankfull was a critical flow to examine. The flood volume (i.e. the cumulative amount of daily discharge above bankfull) is a measure for the extent of flooding and was determined for the modified and the natural flow regimes as mean annual value over the 30-year time period. The percentage change in mean annual flood volume between
modified and natural flow regimes describes the anthropogenic impact on floodplain inundation. Results for the climate change impact are presented as ensemble median, so that the direction of change is reflected by at least 3 out of the 5 selected GCMs. Thereby, only reductions have been documented in the results chapter, because it cannot be distinguished whether an increase in flood volume benefits the wetland or generates flood damages, which, in turn, would be an incentive to build more dams for flood control (Poff and Matthews, 2013). The entire approach described above was carried out for
each single grid cell of the global 5 arc minute raster. For the analysis of riparian wetlands associated grid cells on the WaterGAP3 raster were investigated.

No generalizable relationships between flow alteration and ecological impact are available for large-scale assessments. In order to distinguish levels of modification, 'thresholds for potential concern' were applied for the deviation (Δ) in flood volume between the modified and the natural flow regimes (Table 1). These thresholds are based on the '20% rule' likely
indicating moderate to major changes in ecosystem structure and functions (Richter et al., 2011) and initial thoughts from some water resources experts to set a global standard on eFlow requirements (Hoekstra et al., 2011). However, it has to be



considered that already small reductions in flood volume can result in large decreases in extent of area flooded (Taylor et al., 1996; Kingsford, 2000; Tockner and Stanford, 2002).

In general it can be expected that the greater the deviation from natural condition, the greater the expected ecological impact (Poff and Hart, 2002; Magilligan and Nislow, 2005). Quantitative relationships between peak flows and ecosystems are

provided, e.g. by Wilding and Poff (2008) for rivers in the U.S. state Colorado. In their study, riparian vegetation responds by a maximum change of 12% in community composition for each 10% reduction in peak flows. Consequently, a reduction of 40% in flood volume, which indicates a serious modification in our analysis, could lead to a 48% change in riparian vegetation. Stream invertebrates, in turn, respond exponentially. A 40% change in peak flow caused a maximum response of 54% change in invertebrates.

**2.2    The qualitative assessment of threats**

The modification of river flow regimes will continue in the coming decades. In order to evaluate further threats for riparian wetland flooding, a qualitative assessment was conducted considering future dam construction and the capacity to act. Besides climate change, the construction of new dams will further modify flood pulses and thus, put additional pressure on riparian wetlands. Therefore, for each selected site the number of dams was determined from all upstream dam projects over

10 megawatts in capacity that were planned, proposed or under construction as of July 2014 (Petersen-Perlman, 2014). A number of sources were used to build the dataset: the United Nations Framework Convention on Climate Change's Clean Development Mechanisms (http://cdm.unfccc.int), International Rivers, and other organizations' websites known to fund dam construction (e.g., World Bank). The defined impact on riparian wetlands is shown in Table 2.

In practice, different measures are available to counteract flow alteration threats to riparian wetlands. However, their

implementation is not straightforward and depends on the local capacity to act. In order to assess that capacity for each Ramsar site, two sub-indicators were calculated. The first sub-indicator addresses the availability of water for ecological allocations. A high level of water scarcity in the upstream area indicates a high competition for water resources between different water use sectors and reduces the potential to allocate adequate amounts of water for ecological requirements. In particular, flood pulse provisions would require a relatively large amount of water at a specific time of the year. However, in

some regions, water withdrawals alone can have a strong impact on the river flow regime. For example, water withdrawals at the Murray Darling Basin cause that only 36% of the natural flow drains into the sea (Jolly, 1996). Water scarcity was defined in this study according to Hoekstra et al. (2012) and occurs when the blue water footprint exceeds blue water availability. This approach assumes that no more than 20% of total discharge is depleted by consumptive water use to maintain river ecosystem integrity (Richter et al., 2011). Depending on the average number of month per year with water

scarcity in the upstream area, water availability for ecological purposes was determined (Table 3).



The second sub-indicator addresses institutional capacity and distinguishes between transboundary and non-transboundary upstream areas. For the latter one, the sub-indicator depicts whether the country of the site has legal provisions or official recommendations for the establishment of eFlows (=yes) or not (=no). While having a legal provision is no guarantee that eFlows are actually established in practice, enforced or adequate, it is an important first step for setting strategic goals, advocating ecological water requirements with stakeholders, securing planning resources, and promoting eFlow implementation (Le Quesne et al., 2010). The main sources of information for this indicator were OECD (2015), Benítez Sanz and Schmidt (2012), Le Quesne et al. (2010), and the FAO Water Lex Legal Database (FAO, no date).

In transboundary upstream areas the complexity of water management increases and conflicts are more likely. Formal arrangements governing transboundary river basins, in the form of international water treaties and river basin organizations (RBOs), can be particularly instrumental in managing disputes among different stakeholders involved in water resources management. The greater institutional capacity is, the higher is the potential for eFlow allocations. Institutional frameworks can determine targets, responsible authorities, reoperation strategies, reallocation of water shares, monitoring efforts and consequences of assessment outcomes (Le Quesne et al., 2010; Pahl-Wostl et al., 2013). Therefore, for all sites with a transboundary upstream area, the formal transboundary institutional capacity was expressed by the presence of RBOs, at least one relevant treaty, and specific treaty provisions such as water allocation mechanism, conflict resolution mechanism, and flow variability management. For each of these components present at basin-country unit (BCU) level, one point was given, allowing for a score ranging from zero to five. In order to assign to each wetland a value reflecting transboundary institutional capacity in its upstream BCUs, the values for those upstream BCUs were aggregated and weighted based on the contribution of each BCU to the runoff of the total upstream area. An additional point was given in case the country of the site has legal provisions or official recommendations for the establishment of eFlows. The scores were then grouped into 3 classes describing a low, mid, and high legal and institutional capacity (Table 4). All underlying data were obtained from De Stefano et al. (2012) and complemented with data embedded in international RBOs (Schmeier, no date).

# 3 Results

## 3.1 Quantitative analysis

Riparian wetlands depend on natural patterns of inundation. However, flow regimes of most large river systems in the world have been altered due to different anthropogenic impacts with severe consequences for river ecosystems. Comparing current (1981-2010) modified river flow regimes to natural flow conditions, riparian wetlands of international importance with seriously altered inundation volumes can be found on all continents (Fig. 1). Altogether, half (51%) of the 93 selected Ramsar sites are impaired by at least moderately reduced flood volumes. Eight and 29% of the sites, respectively, are even significantly and seriously affected. In our analysis, dams for hydropower generation are the most frequent dam type in





almost one third of the selected upstream areas, followed by irrigation dams in one fourth of the cases. However, regarding only wetlands with seriously modified inundation patterns, irrigation dams are the most frequent dam type in almost half of the cases (48%). This illustrates the crucial role of irrigation as a strong competitor to ecological water requirements.

In general, a high impact by water resource use appears in Europe especially in the south (mainly affected by dams for irrigation) and in the north (mainly affected by dams for hydropower), but also larger rivers that drain into the Black Sea often show serious modifications (e.g. Dnieper, Dniester, and Don rivers). Nevertheless, a high percentage (56%) of the 43 selected European Ramsar sites possess only slightly impacted inundation patterns The seriously impacted sites are Paúl de Boquilobo (#55) and Doñana (#56) in the Iberian Peninsula, Morava Floodplain (#33), Dnieper River Delta (#41), Lower Dniester (#42), and Dniester-Turunchuk Crossrivers Area (#43) close to the Black Sea, and River Luiro Mires (#14) in the far North of Europe. At these sites flood volumes are reduced due to water resource management by more than 40% in comparison to natural flow conditions. Nine of the analyzed sites are located along the Danube River for which slightly (#31), moderately (#44, #48, #50, #51, #52) and significantly (#35, #37, #46) reduced flood volumes were identified. Due to the lower storage capacities of the numerous dams, the Danube River is more affected by fragmentation than flow regulation as shown by Grill et al. (2015). Significantly impaired flood pulses occur as well at the Elbauen (#20) in Germany. In Europe, hydropower dams are the most frequent dam type in 56% of the upstream areas. The seriously impacted sites in the Iberian Peninsula are mainly affected by water management for irrigation.

In North America, seriously modified flooding patterns can be explained by a high number of large dams for various purposes (hydropower, irrigation, flood control, and water supply). However, in northern Canada, Alaska and southern Mexico many river reaches still show only slightly flow modifications. According to our analysis, two of the four selected North-American wetlands receive seriously (Peace-Athabasca Delta, #1 and Cache-Lower White Rivers, #4), and one significantly reduced flooding (Lac Saint-Pierre, #2). The Emiquon Complex (#3) is only slightly affected by water resource development.

In Australia, mainly river reaches of the Murray-Darling basin are characterized by seriously reduced overbank flow events. For the selected Australian Ramsar areas a strong impact was found in our analysis. Six out of seven sites possess seriously modified flooding regimes: Gwydir Wetlands (#89), Macquarie Marshes (#90), Riverland (#91), Banrock Station Wetland Complex (#92), Barmah Forest (#93), and Ord River floodplain (#88). Except the latter one, all of them are located in the Murray-Darling basin and in their upstream areas more than 100% of the annual flow can be stored in reservoirs causing a high impact on flow regulation, which was also found by Grill et al. (2015). Agricultural irrigation is responsible for the highest water withdrawals and irrigation dams are the most frequent dam type in almost all upstream areas. Kingsford (2000) stated that many floodplains at the Murray-Darling basin have turned into terrestrial ecosystems. One of our selected Australian sites is nearly undisturbed, i.e. Kakadu National Park (#87) located in Northern Australia.





In South America, especially many river reaches in the Amazon basin are still in pristine conditions and many riparian wetlands possess only slightly modified inundation patterns with no or only a few large dams in the upstream area. However, seriously modified inundation patterns are existent in South America as well at three of the nine selected Ramsar sites: La Segua (#5) in Ecuador, as well as Humedales Chaco (#12) and Jaaukanigas (#13) in Argentina.

In Asia, hotspots of river reaches with seriously modified overbank flows are located in India, eastern China and the Middle East. Today, China possesses the highest number of large dams, followed by the United States and India (Rosenberg et al., 2000). In our analysis, one third of the 15 selected Asian wetlands are seriously impacted by water resource development: the Volga Delta (#74) in Russia, Hawizeh Marsh (#77) and Shadegan Marshes & mudflats of Khor-Al Amaya & Khor Musa (#78) at the Persian Gulf, as well as Shandong Yellow River Delta Wetland (#75) and Dong dongting hu (#79) in China. At the Volga River, the construction of dams for hydropower and navigation during the Soviet Union-era has substantially altered the natural flow regime, which negatively influences the dynamics of the Volga Delta (#74). Khublaryan (2000) reported that due to river regulation, mean high water flow decreased from 2/3 to 42% of the annual flow in the Lower Volga. The two Ramsar sites at the Persian Gulf (#77, #78) have been heavily affected by water management and abstractions for irrigation. At Yellow and Yangtze River, particularly large dams for water supply and flood protection play a crucial role.

About half of the selected African wetlands are only slightly affected under current conditions. However, one third of the 15 sites are impaired by seriously or significantly altered overbank flow events. The seriously influenced sites due to current dam operation and water use are Embouchure de la Moulouya (#58) in Morocco, Marromeu Complex (#69) in Mozambique, as well as Baturiya Wetland (#61) and Lower Kaduna-Middle Niger Floodplain (#64) in Nigeria, followed by Sebkhet Kelbia (#57) in Tunisia with significantly reduced flood volumes. At all moderately to seriously affected sites, crop irrigation accounts for the highest water withdrawals in the upstream areas and irrigation dams constitute the most frequent dam type (except at Tana River Delta where hydropower dams are prevailing). Hence, especially in Africa measures are required that balance environmental and agricultural water requirements without reducing food security.

In the future climate change is likely to further modify river flow regimes with consequences for riparian wetland inundation. In regard to decreasing flood pulses, two hotspots became obvious in our analysis for the 2050s: (i) Eastern Europe/ Western Asia and (ii) central South America (Fig. 2).

In Europe, climate change is likely to further decrease flood pulses at more than half of the selected European wetlands as indicated by the ensemble median of the five GCM-projections. At 23% of the sites, reductions might be even significantly or seriously. Most of the concerned sites are located in Eastern Europe, i.e. in the Ukraine (#24, #26, #43), Hungary (#32, #36), Slovakia (#34), Moldova (#42) and Romania (#45), but also occur in Spain (#56) and Germany (#29). Thereby, lower Dniester (#42), Dniester-Turunchuk Crossrivers Area (#43) and Doñana (#56) experience already seriously or significantly





reduced flood volumes under current water management practices so that climate change induces an additional threat. In Asia, wetlands affected by reduced flooding under climate change are located in Russia (#73, #74) and Iraq (#77). At the Volga Delta (#74) and Hawizeh Marsh (#77) flood pulses are already seriously reduced under current water management. In Eastern Europe and Western Asia with their continental climate, global warming is likely to cause a reduction in snow cover leading to lower and earlier snowmelt-induced flood peaks in spring (Schneider et al., 2011b, 2013).

In South America, climate change is likely to decrease riparian wetland inundation at all sites located south of the Amazon River, particularly at the currently slightly affected sites Rio Yata (#8) and Rio Blanco (#9) with reductions of more than 40% according to the ensemble median. At Pantanal Matogrossense (#11) flood pulses might be moderately decreased in the 2050s and remaining wetlands (#7, #10, #12, #13) show slightly reductions between 0 and 20%.

At the selected North American and Australian sites, the ensemble median of the climate projections indicates no further reduction in flood volume for the 2050s. An exception to this is the Gwydir River in the upper Murray-Darling basin with significantly reduced flood volumes in the future. The climate change impact is relatively small on the African wetlands. At only 4 sites (#58, #63, #68 and #69) flood pulses are slightly reduced in the 2050s. However, Embouchure de la Moulouya (#58) and Marromeu Complex (#69) have already seriously reduced flood pulses under current water resource management.

## 3.2    Qualitative analysis

New dam initiatives have the potential to further impair riparian wetland flooding in the future. Altogether, new dams are currently planned or under construction in one third of the upstream areas of the selected Ramsar sites (Fig. 3). The highest percentage was found in South America. Here, two-thirds of the sites are likely to be influenced by new dam initiatives. Especially in the Amazon basin, a very high number of dams is planned or under construction likely to affect flooding patterns of the Mamiraua wetland (#6). Further extensive dam initiatives take place at the Parana and Paraguay River basins. These are likely to impair Humedales Chaco (#12) and Jaaukanigas (#13) with already seriously reduced flooding under current water resource management as well as Pantanal Matogrossense (#11) with slightly reduced flooding under current conditions and a moderate impact by climate change.

A very high number of new dams is also planned in Asia with the potential to affect 60% of the selected Ramsar areas. Major dam initiatives are likely to impair especially the wetlands Dong dongting hu (#79) and Shandong Yellow River Delta Wetland (#75) in China, Sundarbans Reserved Forest (#80) in the Ganges-Brahmaputra-Meghna basin, as well as Tram Chim National Park (#84), Middle stretches of the Mekong River (#81) and Bau Sau Wetlands (#83) at the Mekong River basin. The two Chinese wetlands possess already seriously reduced flood volumes under current conditions.

At about half of the selected African sites, dams are planned or under construction upstream, though the number of dams is relatively small in most upstream areas. The highest number occurs upstream of Marromeu Complex (#69) and Lower





Kaduna-Middle Niger Floodplain (#64) with 12 and 5 dam initiatives, respectively. However, these upstream areas are large in size, so that the effect on the wetlands might be not so strong. A smaller number of dam initiatives is taking place upstream of the moderately affected Delta Interieur du Niger (#59) and some slightly affected sites (#62, #66, #68 and #70).

While a high number of dams have been constructed in North America and Australia in the last century, no further dams are planned or under construction upstream of the selected Ramsar sites. The same applies to most parts of Europe. However, at 19% of the European sites quite a high number of new dams could be constructed upstream in the near future. All of these sites are located in the Balkan Peninsula (i.e. in Croatia, Serbia, Bulgaria and Romania) and the dams are likely to have an impact on riparian wetlands in the lower Danube basin. Currently, the concerned sites are slightly (#47), moderately (#48, #49, #50, #51, #52) or significantly (#46) impaired due to water resource management, but the new dam construction has the potential to further diminish inundation.

Different counteractive measures are available to minimize anthropogenic flow regime modifications. However these measures are complex and require that (i) sufficient water is available to satisfy water demands of different water use sectors and (ii) institutional arrangements enabling the establishment of eFlows are in place. Considering these two factors, Fig. 4 depicts for each riparian wetland the capacity to act in each upstream area.

In Europe, eFlow applications (including high flow provisions) might be most challenging in the upstream areas of currently seriously impacted sites. Water availability for ecological purposes was rated medium for Paúl de Boquilobo (#55) and Doñana (#56) as water scarcity occurs on average in 5 months of the year due to high water requirements for agricultural irrigation. Three to four months of water scarcity were found upstream of Morava Floodplain (#33), Dniepro River Delta (#41), Lower Dniester (#42), and Dniester-Turunchuk Crossrivers Area (#43). In most upstream areas of the assessed European wetlands (71% of the cases), formal institutional capacity was found to be high, with the exception of wetlands located in Ukraine, Belarus and Western Russia. Eastern Europe was defined in this study as one hotspot where climate change is likely to reduce flood pulses in the future. The high number of major dam initiatives at the lower Danube basin presents a risk for conflicts among riparians. However, the high institutional capacity in this region might help to avoid disputes and balance anthropogenic and ecological water requirements. Dniepro River Delta (#41) was found to be an area where special efforts may be needed, due to medium water availability for ecological allocations and a low formal institutional capacity.

In North America, the capacity to act appears to be limited for Cache-Lower White Rivers (#4). At this site water availability for ecological allocations is medium (i.e. four months of water scarcity per year) and legal provisions to ensure eFlows still need to be established. In our modeling inundation volumes are seriously reduced under current conditions, but reoperation schemes could be considered especially for flood control dams, which are prevailing in the upstream area. Also non-structural flood control measures could provide opportunities. For example, reconnecting rivers to their floodplains (where





possible) reduces flood-control storages of reservoirs and thus, increases the potential to allocate water for hydropower generation, water-supply or eFlow provisions (Watts et al., 2011).

EFlow provisions are part of the Australian law and have also been defined e.g. for floodplain wetlands of the Murray-Darling basin (Poff and Matthews, 2013). However, at all seriously affected sites water scarcity is often a major issue. This indicates low water availability for ecological allocations, especially for Banrock Station Wetland Complex (#92), Riverland (#91), Macquarie Marshes (#90), and Gwydir Wetlands (#89) with on average 10, 9, 8, and 7 months of the year with water scarcity, respectively, which reduces the capacity to ensure eFlows. Hence, especially here enhancing water use efficiency in different water use sectors could contribute to riparian wetland conservation.

The potential for ecological water allocations is high for all South-American sites except la Segua (#5), where water scarcity occurs on average in 4 months of the year under current conditions. At Mamiraua (#6), Humedales Chaco (#12) and Jaaukanigas (#13), the formal institutional capacity is medium. Regarding the high number of dams planned in their upstream areas, further institutional arrangements would be of importance for their conservation. The establishment of legal eFlow provisions in the national law could be supportive for the conservation of the Bolivian sites Rio Yata (#8), Rio Blanco (#9) and Rio Matos (#10).

In Asia, the two Ramsar sites at the Persian Gulf (#77, #78) possess the lowest capacity to act. Here, on the one hand, water scarcity occurs on average in six to seven months of the year indicating a low potential for eFlow allocations. On the other hand, legal eFlow provisions are missing in the related national water laws of both sites. The upstream area of Hawizeh Marsh (#77) intersects with four countries and presents a medium institutional capacity. For Shandong Yellow River Delta (#75), seriously modified flood volumes under current water management and a high number of dams planned or under construction were identified in our analysis. Here the blue water footprint exceeds blue water availability on average in 6 months of the year in the upstream area, which reduces the potential to consider the water requirements of the wetland. In Asia, the highest percentage of sites without normative eFlow provisions occurs (87% of the cases). Establishing legal eFlow provisions in the related national water legislation could improve the capacity to act for many sites, in particular for the Volga Delta (#74), where climate change is likely to further reduce the already seriously altered inundation volumes. In the Mekong River basin, the presence of the Mekong River Commission contributes to water-related institutional capacity, which could help to negotiate transboundary issues and implement eFlow provisions.

In Africa, Lake Chad Wetlands (#60), Sebkhet Kelbia (#57), Tana River Delta (#65) and Embouchure de la Moulouya (#58) are riparian wetlands with a low potential for ecological water allocations due to six to eleven months per year with water scarcity. Here, flood volumes are moderately to significantly reduced under current conditions. The high competition for water resources at these sites demands for water use efficiency enhancement. So far, legal eFlow provisions are only considered for 20% of all African sites. Except at Tana River delta (#65), they are missing at all sites with moderately to



seriously reduced flood pulses as well as where new dam initiatives are taking place. Thus, the implementation of legal eFlow provisions could be an important first step to increase the capacity to act at these sites. Detailed results for all wetlands are listed in the Supplement.

## 4        Discussion and conclusions

Freshwater demands of an exponentially growing world population, hydropower development as a new source of renewable energy, and projected climate change pose important challenges to the maintenance of riparian wetlands. Since these provide valuable services and are disappearing at an alarming rate, assessing the alteration of ecologically important flood pulses addresses crucial research questions related to environmental, water and flood management. Therefore, this study aimed at identifying hydrological threats to riparian wetlands of international importance, in particular assessing (i) impacts of current water management on overbank flows, (ii) potential impairments of flooding regimes in the future due to new dam initiatives and climate change, and (iii) the capacity to act required to implement counteractive measures such as eFlow provisions.

Currently, the concept of eFlows is transitioning from an era of ecosystem integrity and conservation at single river reaches to a period of globalization, where regional studies are complemented by global water assessments that cover large-scale developments. Main reasons are increasing socio-economic and climatic changes on global scale and the associated pace of ecosystem destruction, but also because more sophisticated global hydrology models are available now (Poff and Matthews, 2013). In the last years, WaterGAP3 has been further improved and constitutes a state-of-the-art global water model that performs its calculations on a daily time-step and on 5 x 5 arc minutes spatial resolution. In addition WaterGAP3 operates now more than 6000 dams, uses dynamic optimization schemes for different dam types, and considers the water infrastructure of larger cities. The described approach should be regarded as a screening tool that systematically identifies hotspots of threatened riparian wetlands, where further hydro-ecological research should be focused on taking into account local expertise of site-specific ecological, social and economic conditions.

Our approach considers the operation of a high number of large dams with a total storage volume of 6200km$^3$. However, results for water resource management impacts should be regarded as an underestimation, as only large dams that are captured by global datasets are taken into account. The aggregated effect of remaining smaller dams has an impact on floodplain inundation as well (Rosenberg et al., 2000). As no global datasets exist that describe specific operation rules or strategies for individual dams, the dam operation module in WaterGAP3 considers generic operation schemes reflecting the main purpose of each dam. It does not acknowledge eFlow provisions that are already enforced in reality. Thus, the performance of our dam module is lower compared to detailed reservoir models using site-specific information. In reality, inundation is also influenced by river construction (e.g. embankment, re-aligning, widening or deepening) and land-use





changes (e.g. deforestation, land drainage, or sealing of large urban areas). All these influences interact with water resource uses and climate change, but are out of scope of this study.

For this study, 93 Ramsar wetlands, which depend on lateral overspill of adjacent rivers, were analyzed. About half of them are facing no or only slightly impacts due to human water resource management. However, according to our simulations almost one third of them are seriously and further 8% are significantly impaired by reduced flood pulses. Seriously affected sites occur on all continents and particularly in Australia, China, Southern Europe and North America as well as at rivers that drain into the Black Sea or the Persian Gulf. Dam reoperation strategies aiming at ecosystem restoration depend on the dam's main operating purpose (Watts et al., 2011). In our assessment hydropower dams were the most frequent dam type in the upstream areas, however, irrigation dams were prevailing in the upstream areas of seriously affected sites. Consequently, notably for irrigation and hydropower dams, innovative and integrative operating rules need to be developed, which maintain global food security and economic benefits, while at the same time releasing eFlows for ecosystem health and biodiversity.

In the future, climate change will further modify seasonal flow patterns. In the 2050s, the average flood volume is likely to be decreased at 41% of the sites due to the exclusive effect of climate change. At 16% of the sites, reductions can be significantly or even seriously (i.e. >30%). In our analysis two spatial hotspots could be identified: Especially in Eastern Europe/ Western Asia as well as in South America below the Amazon River, flood pulses are likely to be reduced under climate change. Applying a different set of climate projections, lower snowmelt-induced flood peaks in spring were also found for Eastern Europe by Schneider et al. (2011b; 2013). In agreement with results of Zarfl et al. (2014), extensive dam construction is on the way in one third of the upstream areas with potential ecological impacts in particular for South American (67% of the sites), Asian (60%) and African (47%) wetlands. Additionally, countries of the Balkan Peninsula in Europe show a high activity in new dam initiatives. We found that a large impact by future dam construction is likely in the upstream areas of wetlands that are located in the basins of Amazon, Parana, Paraguay, Yangtze, Yellow, Mekong, Ganges-Brahmaputra, and Danube Rivers. As a next step, the new dam initiatives shall be implemented in the model to improve future assessments of ecological and human water stress.

Reduced flood pulses can have lasting ecological impacts such as loss of biodiversity, invasion of non-native species (Poff et al., 1997), modification of river food webs (Wootton et al., 1996), salinization of soils (Nilsson and Berggren, 2000), reduced primary and secondary productivity (Tockner and Stanford, 2002), as well as habitat deterioration and loss of floodplain wetlands, river deltas and ocean estuaries (Rosenberg et al., 1997; Rosenberg et al., 2000). For the identified hotspots of current and future threats, the implementation of appropriate eFlows is likely to be most urgent. However, this is a complex task and requires a high capacity to act which depends in particular on two factors: First, the degree of water resource competition in a river basin determines the amount of water that can be allocated for ecological purposes. In our analysis, the highest competition for water is existent in the upstream area of Lake Chad Wetlands (Nigeria) followed by wetlands of the river basins Murray-Darling (Australia), Schatt al-Arab (Persian Gulf), Tana (Kenya), Moulouya (Morocco),





and Yellow (China), where water scarcity occurs upstream on average in six to ten months of the year. Therefore, especially at these sites, measures are required which increase water use efficiency (e.g. by water recycling, technological innovations, dripping irrigation, changing crop mix, importing agricultural products, water metering or other incentives to save water) to raise the amount of water which can be allocated for ecological requirements. Second, legal and institutional capacities must be in place that promote stakeholder involvement, conflict resolution, monitoring as well as setting of strategic goals and responsibilities (Le Quesne et al., 2010; Pahl-Wostl et al., 2013). A first important step for eFlow implementation is the acknowledgement of ecological water requirements in the legislation. Even if this does not guarantee that eFlows are actually established in practice, enforced or adequate, it shows that ecological water requirements are on the radar of legislators and water practitioners, and helps advocating for ecological water requirements. At about half of the sites normative eFlow provisions are considered in the national or state Water Act of the wetland. The highest percentage of sites without normative eFlow provisions occur in Asia (87% of the cases) and Africa (80%).

The more countries depend on the available water resources and affect the river flow regime with their management, the more challenging is the implementation of eFlows. About half of the 93 selected wetlands have transboundary upstream areas and the highest percentages appear in Europe (65%) and Africa (60%). For all transboundary upstream areas it is important that river basin organizations, international water treaties and specific treaty provisions are in place that manage disputes and water resource allocation among different stakeholders. The lowest values for formal institutional capacity became obvious in our analysis in the transboundary upstream areas of wetlands that are located in the Ukraine, Belarus and Russia. A medium institutional capacity in combination with new dam construction occurs in the transboundary upstream areas of Mamiraua (#6), Humedales Chaco (#12) and Jaaukanigas (#13). At these sites, the establishment of formal arrangements would be of special importance for riparian wetland conservation. While the institutional capacity indicator considers national laws, international treaties and RBO agreements, it is important to stress that the presence of formal arrangements is no guarantee that they are effectively enforced in practice. Altogether, the lowest capacity to act was found for Sebkhet Kelbia (#57), Embouchure de la Moulouya (#58), and Shadegan Marshes (#78).

Climate change and growing population pressures ask for immediate action to conserve wetlands and river ecosystem integrity. The concept of eFlows can be an important strategy for sustainable development and offers opportunities for society as a whole to benefit from vital ecosystem services of riparian wetlands such as maintenance of genetic diversity, production of food, fiber and fodder, decomposition of pollutants and nutrients, provision of local recreational areas and tourism economies, and control of devastating flood events. In practice, the provision of eFlows for wetland inundation is associated with some challenges. In particular it needs to be assured that high-flow pulses do not expose people to flood risk and damage. This may require, for example, defining a maximum admissible flow for river reaches, buying land from farmers, or establishing floodways that direct floodwater around human settlements. Today, eFlows are defined at only a tiny fraction of rivers worldwide and in most cases those are restricted to low flows (Richter et al., 2011). The potential to





implement eFlow provisions and exploit opportunities is by far not exhausted. Even synergies rather than trade-offs between sectors might be possible.

## Acknowledgements

Research funding for this study was partially provided by the 'Global Environment Facility Transboundary Water Assessment Programme' (GEF TWAP, http://twap-rivers.org/) and is gratefully acknowledged. Data on riparian wetland location and extent were downloaded from the website of the Ramsar Convention Secretariat ('http://www.ramsar.org/').

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



**Table 1 : Thresholds for different levels of mean annual flood volume deviation (Δ) between modified and natural flow regimes (as suggested for global assessments by Hoekstra et al., 2011)**

| River status | Level of modification | Thresholds for reduction in flood volume |
|---|---|---|
| A | not/ slightly | $\Delta \leq 20\%$ |
| B | moderately | $20\% < \Delta \leq 30\%$ |
| C | significantly | $30\% < \Delta \leq 40\%$ |
| D | seriously | $\Delta > 40\%$ |



**Table 2 : Defined impact on a riparian wetland subject to the number of new dam initiatives in the upstream area**

| Number of major dam initiatives | Potential impact |
|---|---|
| 0 | NONE |
| 1 – 12 | MED |
| 28 – 276 | HIGH |





**Table 3 : Water availability for ecological allocations defined by means of the number of month with water scarcity upstream of the Ramsar site.**

| Number of month with water scarcity | Water availability for ecological allocations |
|---|---|
| 6 – 12 | LOW |
| 2 – 5 | MED |
| 0 – 1 | HIGH |



**Table 4 : Formal institutional capacity in transboundary upstream areas of Ramsar sites**

| Score | Institutional capacity |
| --- | --- |
| 0 – 2 | LOW |
| 3 – 4 | MED |
| 5 – 6 | HIGH |





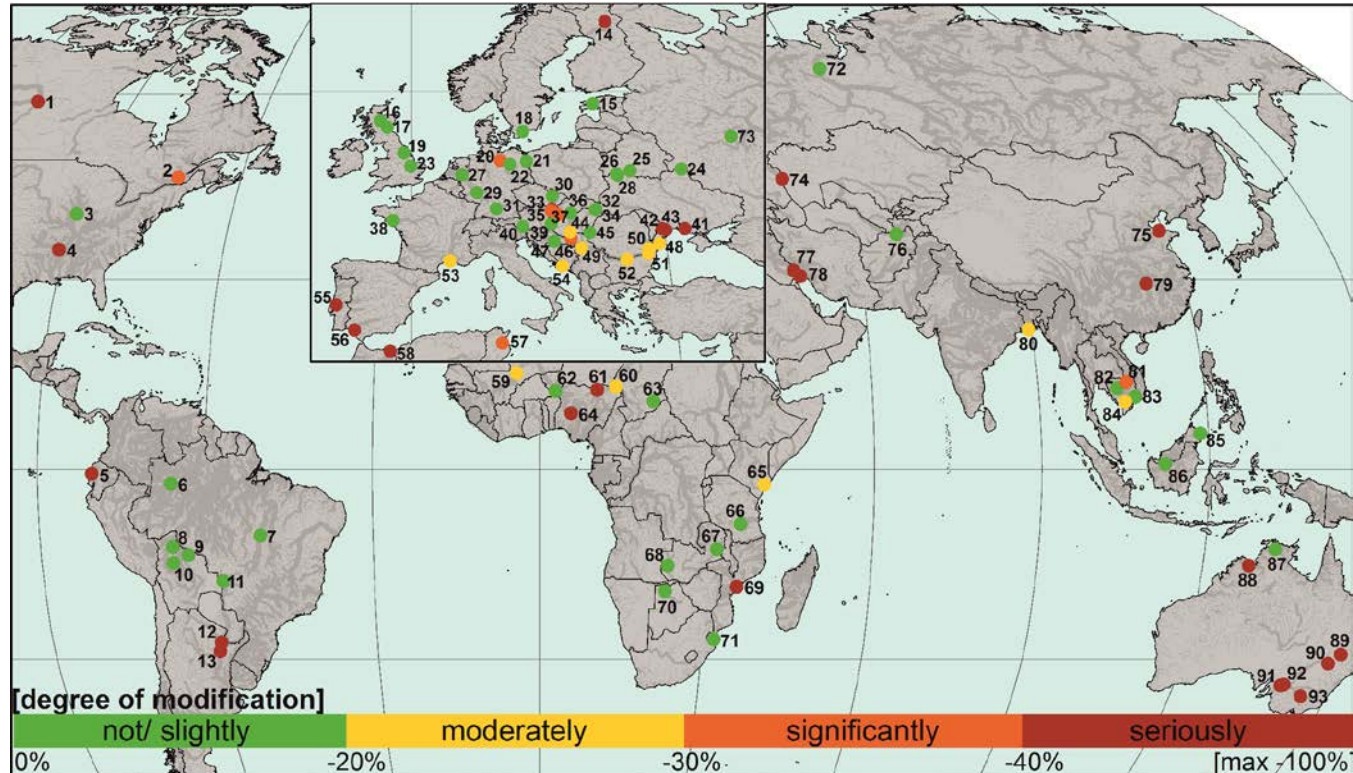

**Figure 1: Global map of overbank flow alterations for selected riparian wetlands of international importance (#1-93) as a consequence of current water resource management**



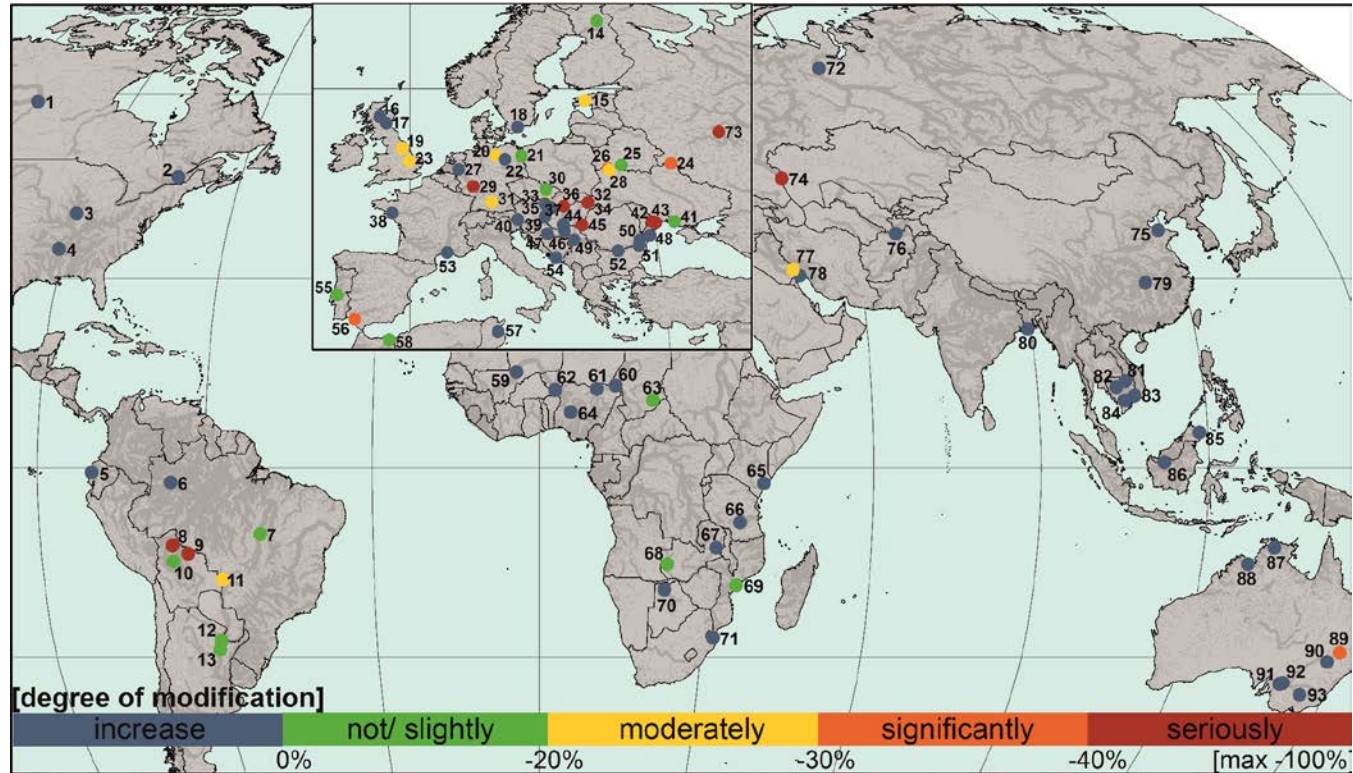

**Figure 2: Global map of overbank flow alterations for selected riparian wetlands of international importance (#1-93) as a consequence of the exclusive effect of climate change in the 2050s.**



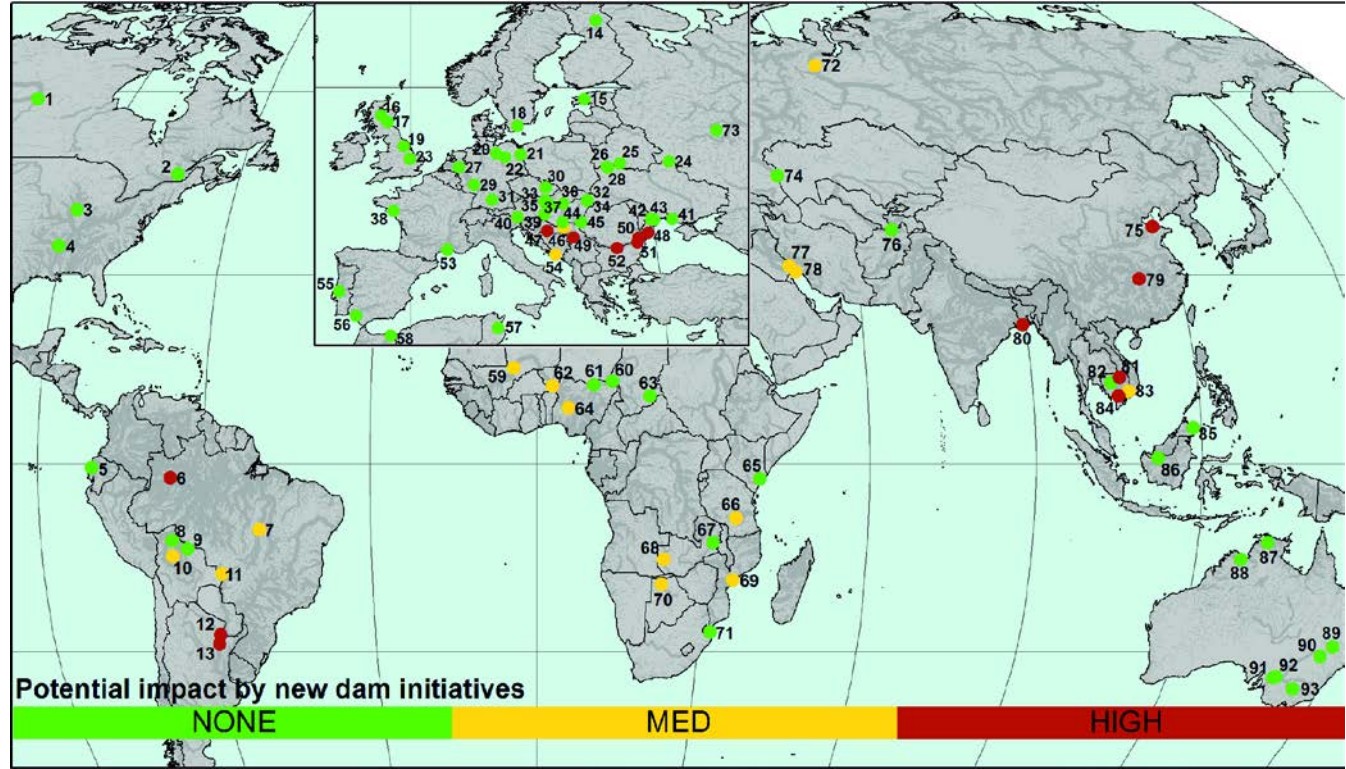

**Figure 3: Potential impact by new dam initiatives taking into account the number of dams currently planned or under construction in the upstream area of each riparian wetland.**





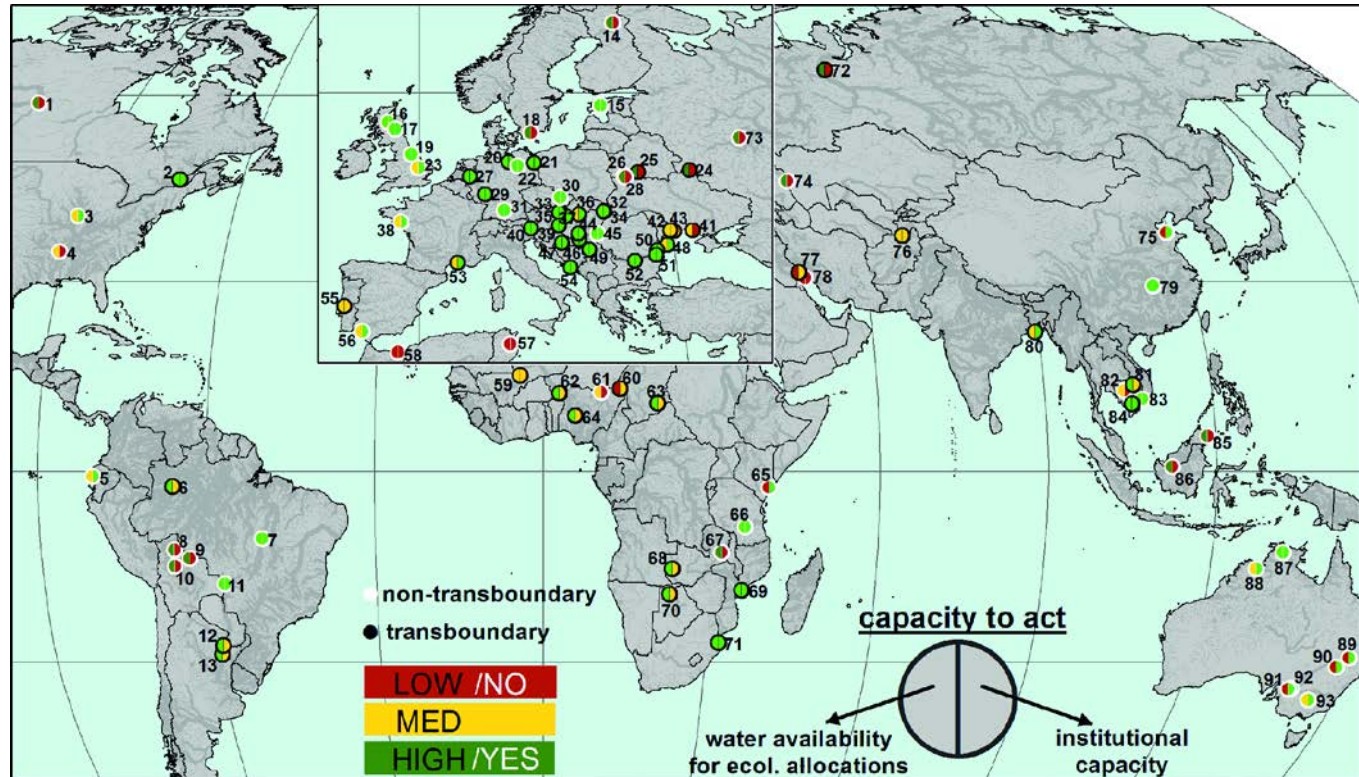

**Figure 4: Current capacity to act in regard to anthropogenic flow regime modifications for selected riparian wetlands. The left semicircle represents the water availability for ecological allocations, while the right semicircle characterizes the institutional capacity in the upstream area. For wetlands with a non-transboundary upstream area (white border), the right semicircle**
5    **represents only presence or absence of legal provisions or official recommendation to establish eFlows.**