# Peer review of "Hydrological threats for riparian wetlands of international importance – a global quantitative and qualitative analysis"

_Hydrology and Earth System Sciences, 2016_

## Referee Comment (RC1) · Anonymous Referee #1 · 23 Aug 2016

Anonymous Reviewer

**August 22, 2016**

Natascha Töpfer
Copernicus Publications
Editorial Support

RE: Review of the paper titled, "Hydrological threats for riparian wetlands of international importance – a global quantitative and qualitative analysis"

Hello Ms. Töpfer.

Thank you for the opportunity to provide an internal review of the paper *Hydrological threats for riparian wetlands of international importance – a global quantitative and qualitative analysis.* The authors have created an innovative and useful screening tool to flag particularly vulnerable wetlands. I am particularly impressed by the integration of quantitative and qualitative methods to achieve their goals. This will be a strong contribution to the literature and useful to organizations seeking to target wetland conservation funds. In addition, the same schematic approach can be applied far beyond wetlands, further enhancing the potential impact of this work. Below, I discuss the major suggestions I have for the manuscript. Finally, I have added comments well over 100 comments directly to the PDF version of the paper and figures, included below within this same document. To read my embedded comments, please hover your cursor over the many red and orange colored icons, shaped like keys, stars, and text bubbles.

1) I feel the authors need to very clearly acknowledge the essence of their study. The way I read the paper, the authors created a diagnostic tool, or procedure, to flag wetlands *likely to be* most vulnerable in the future. It bridges quantitative and qualitative research, applied in a high profile setting. Most importantly, it helps solve the problem of scale in global assessments. This product satisfies the stated goals (p. 4, line 29 through p. 5, line 7). This is a product worthy of praise and significant to both the science and management communities.

Unfortunately, the study is not presented in this light. This sets the stage for the real problem with the paper, that the authors do not consistently or explicitly discuss the limitations of the study, resulting in chronic problems with overstating conclusions - addressed in my second major comment.

To remedy the situation, I recommend:
    1. Reframe the study as the development of a screening tool
    2. In the Discussion/Conclusions section, refrain from restating what the results are, and instead focus on what the results mean. Specifically that they guide future research and/or allocation of resources for wetland protection
    3. In the Discussion/Conclusions section, add a very clear explanation of the study limitations.
    4. Avoid making conclusions about the wetlands themselves - other than to say they are *likely to be* vulnerable or not. Or that they are vulnerable *under future simulated conditions*. Perhaps discuss why hotspots exist. There absolutely is no basis to prescribe specific management action other than where to direct conservation resources and future investigation.

2) I feel the authors overstate their conclusions. However, this appears to be a symptom of a structural problem with the paper. With regard to the Methods section, the authors made many decisions on how to go about their study. This is inherent in any study and particularly messy when trying to scale up to a global assessment. Unfortunately, very little is said about the rationale for their decisions. I'm not saying I think the authors made poor decisions; I'm merely saying they should share their rationale. For example, they chose WaterGAP3 as a model. But they never state why they chose that model. I would like to see that they actually thought about other options and felt WaterGAP3 was the best for some actual reason. Also, the authors chose one particular climate change projection to use in their 50-year forecast. I'm happy they discuss this scenario as one of many. But there is no rationale provided for why they chose this one. They could add sentence and I would be happy. I flag several instances of this in my embedded comments.

   More importantly, I would like to see the Discussion section evaluate the consequence of these decisions, if relevant. For example, I wonder how sensitive the results are to using that one specific climate projection. Or not considering present allocations of eFlows. This helps delineate the limitations of the study and helps guide where future study should focus.

   Once the authors have thought through and delineated the limitations of the study in the Discussion section, they will be far less likely to overstate the conclusions.

3) I feel the manuscript could better "funnel out" in the Conclusion section. This is a chance for the authors to wave their flag and tell me why their work is important. Scientifically, I'm impressed that they combined qualitative and quantitative methods to navigate issues of scale in a global assessment. They should feel free to state this as an academic contribution. The implications of this study could be far reaching. Guiding resources to protect Ramsar wetlands is a big deal. And sure, this study has its limitations. But the authors created a template that could have more broad applications. This study focused on Ramsar wetlands. But maybe the next one could be a true global wetland assessment. Maybe this template could be tweaked and applied to settings such as coral reefs, forest production, or water supply. I wish the authors would express a vision for how this study advances us in the big picture.

4) My one objection to their methods is the throwing out of results that suggested *more* overbank flooding will occur in 2050 than occurs now. This *might* be justified, but it sounds fishy to me. Of course climate change will cause flooding to increase in some places and decrease in others. Why wouldn't they include that in their assessment? At the very least, I recommend this decision be discussed and a rationale provided. Not providing other studies that have done similarly makes me suspicious this isn't a valid assumption. The Discussion section should provide some assessment of how this decision affected the results.

5) I sense the paper was drafted by a non-native English speaker. I am supportive of this and welcome different perspectives in the literature. Unfortunately, I had a difficult time understanding the content. If left unaddressed, I feel this will reduce the impact of the paper. I have some specific recommendations to move forward.
   1. Adding subsections would greatly help keep the text organized.
   2. Please keep paragraphs short and focused on the topic sentences. For example, the first paragraph of the introduction is almost a page long and drifts away from the topic sentence. That is too much.

3. Adding a flow chart to the methods section that schematically illustrates your 3 modeling exercises would be very effective at communicating what you did. I visualize your methods as having 3 'cuts' of modeling: "natural" conditions, "natural + *modeled* water management," and "climate change 2050 without water management." Even if/when readers become confused by the text, a nice flow chart will communicate your general approach well. See my embedded comments.

4. I feel adding a table to the Results section is essential and will allow you to delete at least half the text in the current Results section. I visualize one row per wetland in the study and one column each for wetland number, wetland name, vulnerability for the three conditions tested, and perhaps a comment column. The Results section text could be reserved to identify trends and hotspots rather than telling the reader verbally what wetlands were vulnerable. To be clear, any text that simply tells the reader "wetland X was vulnerable" could be deleted and replaced with more substantive information.

5. The Discussion section is largely dedicated to re-presenting results. In my own writing, I typically find this to be my #1 problem. Deleting any presentation of results in the Discussion section will free up vast amounts of text to focus instead on what the results mean to your study goals and the limitations of your results.

6. Similarly, conclusions are also presented in the Results section and this text should also be removed. I flag these instances in my notes on the manuscript PDF, included below within this same document.

7. Other issues exist, notably sentence structure. My sense is putting the paper through an editorial review would be the most expedient solution. There is a lot that a good editorial reviewer can add that I simply cannot.

To conclude my thoughts on the manuscript, I feel it is publishable with major corrections. The corrections, however, are largely limited to the presentation of the paper, not a fundamental problem with the methods, per se. Again, I have added ~130 comments to the manuscript PDF, included below, to further guide revision of the manuscript. Thanks again for the opportunity to review this paper.

Anonymous Reviewer

[revised manuscript text omitted]

Additional levels of organization would be helpful.

the 93 selected

**2.1 The quantitative assessment of threats**

[Figure]

In order to quantitatively assess anthropogenic alterations of flood pulses at Ramsar sites, the following procedure was taken: (i) simulation of daily natural flow regimes for the time period 1981-2010, (ii) simulation of daily flow regimes for the same time period modified by current water resource management, (iii) simulation of daily flow regimes for the 2050s according to climate change projections, (iv) estimation of bankfull flow as a crucial threshold that marks the starting point of inundation, (v) analysis of all overbank flows by calculating the mean annual flood volume for the modified and the natural flow regimes, and (vi) determining the deviations in flood volume in the modified flow regimes in comparison to natural flow conditions.

For the simulation of daily river discharge, the integrated WaterGAP3 modeling framework was applied (Verzano, 2009), which performs its calculations on a global 5 x 5 arc minute grid cell raster (~9 x 9 km$^2$ at the Equator). The global hydrology model of WaterGAP3 computes the macro-scale behavior of the terrestrial water cycle. In order to run it for the time period 1981-2010, the WATCH-Forcing-Data-ERA-Interim (WFDEI; Weedon et al., 2014) were used as climate input. It consists of a set of daily, 0.5 x 0.5 degree gridded meteorological forcing data, which were simply disaggregated to the 5 arc minute resolution as required by the model. Forced by the climatic time series, WaterGAP3 calculates daily water balances for each grid cell taking into account distributed physiographic characteristics from high spatial resolution maps describing slope, soil type, land cover, aquifer type, permafrost and glaciers, as well as extent and location of lakes and wetlands. The total runoff in each grid cell, derived from the water balances of land and freshwater areas, is routed along a predefined drainage direction map (DDM5; Lehner et al., 2008) to the catchment outlet. Land cover data were derived from the Global Land Cover Characterization map (GLCC; USGS, 2008) and for EU countries from the CORINE Land Cover map (CLC2000; EEA, 2004). This entire setting was used to gain near-natural flow regimes.

[Figure]

For the flow regimes modified by current water resource management, additionally, anthropogenic flow alterations due to water use and dam operation were taken into account. For these model runs, the natural discharge was reduced in each grid cell by consumptive water use as calculated by the global water use models of WaterGAP3. These models simulate spatially distributed sectoral water uses for the five most important water use sectors: electricity production, manufacturing, domestic use, agricultural crop irrigation and livestock watering (Aus der Beek et al., 2010; Flörke et al., 2013). Assuming an optimal water supply to irrigated crops, net irrigation requirements are simulated for each grid cell based on climatic conditions, dominant crop type and irrigated area around the year 2005 (GMIAv5; Siebert et al., 2013). Livestock water demands are determined by multiplying the number of animals per grid cell by the livestock-specific water use intensity (Alcamo et al., 2003). The amount of cooling water consumed by the electricity production sector is calculated by multiplying the water use intensity of each power station with the equivalent annual thermal electricity production. The water use intensity is affected by the cooling system (once-through flow cooling, tower cooling, or ponds) and the type of fuel (coal and petroleum, natural gas and oil, nuclear, or biomass and waste) used at each power station (Flörke et al., 2012). Power station characteristics

Paragraph, selected at random, to show just how difficult the paper is to follow.

[revised manuscript text omitted]

How about a caption:
"Magnitude of modeled water-resource management alteration to overbank flow near 93 riparian wetland study sites."

[Figure]

[Figure]

**Figure 2: Global map of overbank flow alterations for selected riparian wetlands of international importance (#1-93) as a consequence of the exclusive effect of climate change in the 2050s.**

[Figure]

[Figure]

**Figure 3: Potential impact by new dam initiatives taking into account the number of dams currently planned or under construction in the upstream area of each riparian wetland.**

[Figure]

[Figure]

[Figure]

**Figure 4: Current capacity to act in regard to anthropogenic flow regime modifications for selected riparian wetlands. The left semicircle represents the water availability for ecological allocations, while the right semicircle characterizes the institutional capacity in the upstream area. For wetlands with a non-transboundary upstream area (white border), the right semicircle represents only presence or absence of legal provisions or official recommendation to establish eFlows.**

---

## Referee Comment (RC2) · Anonymous Referee #2 · 19 Oct 2016

Review of Schneider et al "Hydrological threats for riparian wetlands of international importance – a global quantitative and qualitative analysis"

Through data synthesis and model interpretations of RAMSAR wetland sites across the world, this paper addresses the issue of past to expected future adverse effects on riparian wetlands from pressures such as climate change and water regulation. In particular the focus is on the available flooding volume - how it has been modified today and how it may change in the future due to these pressures. The magnitude of these changes are taken as a measure of potential ecological impacts.

The authors combine and use multiple methods (e.g. to simulate impact of flow regulation of various dam types etc), many of which have been thoroughly developed in

previous work. Although results are associated with considerable uncertainties, the approach is quite reasonable and the outcome is logically synthesised and presented as maps showing e.g. the magnitude of flow alteration impact. Such global state-of-the-art syntheses is certainly of scientific interest; I would recommend publication of the work if main shortcomings (see below) can be addressed, which is likely to require at least moderate revisions.

In summary, these shortcomings are (1) lack of clarifications regarding novel aspects of the present study, apart from the novel global synthesis perspective, (2) partial lack of information regarding past experiences of the proposed methods, (3) language issues, (4) lack of sufficient results comparison to previous studies, and (5) unfocused conclusions. Overall, this study has high potential and I hope that the detailed comments below can be useful in addressing the current concerns.

1. Presently, the focus of the introduction is on the relevance of the topic, including what is known about vital ecosystem services of floodplain wetlands, effects of dams in a more general sense, and the need for maintaining flow variability etc. This description is on the lengthy side and could probably be condensed. However, more concrete (state-of-the art) regional examples that presumably exist in the scientific literature regarding today's impacts (or expected future impacts) on floodplain wetlands are essentially missing. Such examples should be included in the introduction, such that the readers can understand what is novel about the presented result-maps, in addition to the novel global synthesis perspective. In other words: which previous indications exist in the scientific literature regarding key results, such as the result showing that the degree of overbank flow alteration due to current management is very low in Europe (essentially green in Figure 1) whereas Australia comes out as seriously altered (or other results that are the authors think is important). I would recommend the authors to go through what they consider to be the main results of their study and make sure that the introduction informs sufficiently about the current knowledge. This would provide a necessary basis for enhancing the discussion (see bullet point 4)

2. It is stated in the introduction (p. 2, line 26) that a new approach is needed to water resources management, which among other things should allow for sufficiently high flows for sustaining floodplain wetlands. However, in line with comments of bullet point 1 (above), this proposed novelty remains unclear to the reader. For example, haven't we gained some relevant knowledge from regulation schemes applied to the principal Colorado River in the US (Stevens et al., 2001; Stromberg et al., 2007; Cross et al., 2011)? These schemes have included controlled floods as part of the strategy to minimise adverse impacts to downstream ecosystems. Perhaps there other relevant examples.

3. The language of the manuscript is overall good. There are some exceptions though, including the introduction. In particular, the research questions and the related text include awkward formulations (e.g., multiple sentences starting with Thereby.../ Therefore...), please check.

4. There is a lack of results comparison to previous studies in the discussion section, which should be addressed before publication. The now included references do mainly not relate to the results (study outcomes) and need therefore to be complemented. For instance, are the results regarding impacts on the 93 Ramsar wetlands in different world regions (p. 17, lines 3-11) consistent with previously reported results for these regions? Alternatively, do the results partly contradict or point to new and previously unnoticed aspects? (Also, the reader is not well informed about the existence or absence of similar studies, see bullet point 2 above regarding the introduction). The same questions can be asked for other key results, such as impacts of climate change and the related identified hotspots (p. 17, line 12-15), and competition of water (p. 17, lines 31-32). Overall, the discussion section is rather general and would benefit from an extended discussion of results. The aims of the study need not to be reiterated in the beginning of the discussion section.

5. The main conclusions of the paper are not clearly presented. Maybe a separate conclusion section could help?

References

Cross, W. F., Baxter, C. V., Donner, K. C., Rosi-Marshall, E. J., Kennedy, T. A., Hall, R. O., ... & Rogers, R. S. (2011). Ecosystem ecology meets adaptive management: food web response to a controlled flood on the Colorado River, Glen Canyon. Ecological Applications, 21(6), 2016-2033.

Stevens, L. E., Ayers, T. J., Bennett, J. B., Christensen, K., Kearsley, M. J., Meretsky, V. J., ... & Springer, A. E. (2001). Planned flooding and Colorado river riparian trade‐offs downstream from Glen Canyon dam, Arizona. Ecological Applications, 11(3), 701-710.

Stromberg, J. C., Beauchamp, V. B., Dixon, M. D., Lite, S. J., & Paradzick, C. (2007). Importance of low‐flow and high‐flow characteristics to restoration of riparian vegetation along rivers in arid south‐western United States. Freshwater Biology, 52(4), 651-679.
* * *

---

## Referee Comment (RC3) · Anonymous Referee #3 · 27 Oct 2016

The authors aim to address an important issue: Identifying Ramsar riparian wetlands that exhibit current and future variations in ecologically consequential inundation patterns as a result of human-modified flows (e.g., dams). They ask three particular research questions to best identify these wetlands. These questions focus on the impact of current water resource management on riparian wetland flows, the effect of future climate change on inundation of these wetlands, and the implications of low government and societal infrastructure and capacity to make changes to future management.

The goal and research questions the authors attempt to address are broad and could be impactful if addressed and translated well. However, a major revision is required to ensure both the quantitative work behind the research and the communication of this

work is effective. Below, I provide major suggestions for the manuscript followed by some general comments.

Major Point 1: The Introduction reads somewhat like a full literature review that continues for quite some time without a direct point. It was well into the sixth paragraph that the goal and research questions were stated. I would suggest tightening up the Introduction, providing only key points throughout, and early on (perhaps at the end of the first paragraph) allude to the main point of the paper (e.g., "We aim to. . ."). Then, the authors can safely state the full objective and research questions at the end of the Introduction.

Major Point 2: Something is very misleading and incorrect about discussing a "natural" flow regime in knowingly modified watersheds and aquatic systems. Also, the word "natural" is used throughout the Abstract and Introduction (ala Poff et al. 1997), and it is not until the Methodology that authors define natural flow. The authors describe natural flow for this paper as "simulated taking into account current climate and land-cover conditions, but no further anthropogenic impacts." This, by no means, would constitute a "natural" flow regime as described in past literature. I would recommend modifying terminology and the discussion throughout the paper to consider this as your "baseline" flow regime from which the analyses aims to understand current water resource management implications on the riparian wetlands and project changes of these regimes due to climate change

Major Point 3: The goals of the paper and research questions are poorly worded need more information. What, specifically, are the "riparian wetlands?" In the Abstract, the authors suggest they look at 93 Ramsar sites. Are the "riparian wetlands" the "93 Ramsar riparian wetlands?" For Research Question 1, why are 6025 dams selected? Are these dams specifically located upstream of Ramsar riparian wetlands? What are the "different water use sectors"? Also, delete "Thereby" at the beginning of the second sentence. For Research Question 2: "Inundation" cannot be "impaired" because "inundation" does not necessarily denote a positive quality. The authors could replace
"impaired" with "exacerbated or diminished" or "modified." Also, delete "Therefore" at the beginning of the second sentence. Research Question 3 is stated in a grammatically incorrect way, so it took a few re-reads to understand it. Move "could" after the word "sites." Also, what is a "low capacity to act?" This is definitely not clear.

Major Point 4, WaterGAP3 runs: Streamflow, for what the authors term "daily natural flow regimes" (1981-2010), is simulated with 2004 land cover. Using 2004 land cover is okay; however, going back to the use of the word "natural"...how can this be considered natural flow when the landscape for each area is likely highly modified and streamflow is a reflection of these anthropogenic activities? Also, there is no mention of calibration and verification of the model, which admittedly would be difficult a global scale. Therefore, is the entire paper a thought experiment using an uncalibrated global model to help explore hypotheses? It would be okay if so, though this framework should be characterized as such throughout the paper. Also, the results (maps, in particular) should emphasize the paper's overarching approach (i.e., the thought experiment – a "screening tool" is mentioned in the Discussion, hypotheses testing, and/or a conceptual model). If calibration and verification did occur at some stage and is not referenced, again, measured streamflow would reflect the managed conditions, not some unattainable "natural" or "near natural" condition. The model scenarios are therefore a bit confusing and need some rethinking, definitely in the presentation of what they are but potentially in which ones should be used. For example, consideration should focus on whether only the managed scenario and future climate/management conditions should be used since the true "natural flow regimes" aren't captured.

Also, the authors talk about the database of dams that are used, but how does that relate back to the Ramsar wetlands? Are these dams all upstream of Ramsar wetlands? As I read on, it became a bit clearer that this is simply a global database, and Ramsar wetland areas within the global domain are analyzed. However, this information (spatial domain and selection of dams) needs to be clearer up front.

The authors likely have all the information mentioned in this Major Point. There is

simply a need for better and clearer communication regarding these bits of information. As a result, the Methodology section seems quite disjointed and leaves the reader guessing at how the authors conducted the analyses.

Major Point 5, Discussion and Conclusions: Be careful here. Because this is thought experiment using a global model (again, unless calibration/verification happened but wasn't mentioned), your conclusions need to be balanced with a statement of the conceptual aims of the paper and associated limitations/assumptions. The quantitative analyses isn't incredibly quantitative, and I wince a bit with the use of numbers like "8 % are significantly impaired" and flood volume is likely to be decreased at 41% of the sites. . ." when those are all relative numbers with no basis in reality. Please mention up front in the conclusions or make a separate section of the limitations and assumptions with regard to what the analyses can actually provide.

Major Point 6: In general, the English is okay as written. However, it's important that someone extremely proficient in English re-review this paper for odd placement of verbs, adjectives, modifiers, etc., and poor word selection. One small example, on Page 5, Line 14 "For Europe, a higher number of sites "were gained" as the European wetland geodatabase. . .". This should be "were selected" or "were chosen". There are many instances like this throughout the paper, and I do not list them all below.

Specific Comments

Page 1, Line 9 – Recommend changing all references of "mankind" to "humankind" and "man-made" to "constructed"

Page 1, Line 9 – These eco services are provided not only via the regular patterns of inundation but also regular patterns of drying – so actually, it's the *variability* inundation patterns that is important.

Page 1, Line 26 – Need to review and add Dixon et al (2016) as well. Dixon, MJR, Loh J, Davidson NC et al. 2016. Tracking global change in ecosystem area: The Wetland

Extent Trends index. Biol. Conserv. 193: 27-35.

Page 2, Lines 18-19 – Is this true for all "larger cities?" What spatial scale is this referring to? Are these global or regional estimates? If regional, what regions?

Page 2, Line 25 – Again, what are "natural sites?"

Page 2, Line 30 – Not all floodplains are wetlands, which is how this sentence reads. Please correct.

Page 2, Line 32 – What ecological processes are initiated? Some of these processes may be initiated by drying not wetting.

Page 3, Line 2 – What is engendering what? This clause doesn't make sense.

Page 3, Line 4 – That's a very broad statement, that all floodplain wetlands contain more species than any other landscape unit. Need more specifics here because it's likely not what the authors intended to say.

Page 3, first paragraph – The Roman numerals are not needed when providing full sentences after them. Suggest removing all Roman numerals here.

Page 3, Line 24 - What are "fellow riparians?" Please be more specific.

Page 3, Line 25 – What projections? Please be more specific.

Page 4, Lines 12-16 – Break up this sentence into two or more sentences.

Page 5, Lines 19-20 – These sentences can be deleted and are unnecessary.

Page 5, Line 23 – "percent change in flood volume": from what period to what period? Please provide time frame.

Page 5, Line 28 – It is not clear at this point what "sufficient capacity to act" means. Suggest modifying this or adding some clarification here to lead the reader to the more specific methods discussion.

Page 5, Line 3 – The simulation of daily natural flow regimes would still be an expression of a modified landscape, so how are these natural?

Page 5, Lin 9 – Need clarification of what type of "daily river discharge" is being simulated here – "natural" or "managed"? (After reading on, it becomes obvious it's "natural" but that needs to be mentioned straight away.)

Page 5, Line 20 – Switched to "near-natural" from "natural" in this sentence. Please be consistent.

Page 7, Lines 6-9 – Need to be clear here why the simulation includes these specific 6025 dams. Why were they chosen? Intuition would tell me they are all upstream of Ramsar sites, but further reading seems to suggest that they are simply part of the global database. These questions regarding methods also suggest that clear summary statements of what the quantitative analyses is up front in the Methodology should be added – meaning state your steps: exact simulations, the spatial scale, how dams were selected, how the Ramsar sites were overlain on the global map, etc. Then, details can be added after this summary.

Page 9, line 14 – What selected sites? The Ramsar wetlands? Again, details are needed here.

Page 9, Lines 14-15 – This sentence is a bit wonky and needs to be reworded.

Page 9, Line 18 – How were the cutoff thresholds for Table 2 selected?

Page 9, Line 20 – Again, clarify what the "low capacity to act" is.

Page 9, Line 27 – Define blue water

Page 9, Line 30 – Again, how were the Table 3 thresholds derived?

Page 10, Lines 25-27 - Cut these sentences. Too much introduction here.

Page 11, Line 4 – These wetlands are "moderately impacted" – as far as the map

seems to read.

Page 11, Line 17 – N=2, though, correct? So this is only discussing two wetlands, right?

Page 12, Line 28 – Is this the ensemble median for the GCMs as input to the Water-GAP3 model or the ensemble average of the output of the WaterGAP3 model?

Page 13, Liens 16-17 – Now that is a very interesting finding!

Table 1, change "not/slightly" to "none/slightly" – same with the figures: "not/slightly" does not make sense.

Table 2, delete "the number of" in the caption.

Table 4, define "formal institutional capacity" in the caption to make the table stand alone.

The final edits for the paper are included in the Major Points listed previously.

---

## Author Comment (AC1) · 24 Nov 2016

**Response to Referee #1**

*We thank Referee #1 for the profound and detailed evaluation of the paper and the helpful comments, which will further improve this paper.  We are confident that we can adequately address each of these comments and understand that we need to work especially on the presentation of our results. Please find below our responses describing our planned revisions (highlighted in blue and italic type).*

Hello Ms. Töpfer.

Thank you for the opportunity to provide an internal review of the paper Hydrological threats for riparian wetlands of international importance – a global quantitative and qualitative analysis. The authors have created an innovative and useful screening tool to flag particularly vulnerable wetlands. I am particularly impressed by the integration of quantitative and qualitative methods to achieve their goals. This will be a strong contribution to the literature and useful to organizations seeking to target wetland conservation funds. In addition, the same schematic approach can be applied far beyond wetlands, further enhancing the potential impact of this work. Below, I discuss the major suggestions I have for the manuscript. Finally, I have added comments well over 100 comments directly to the PDF version of the paper and figures, included below within this same document. To read my embedded comments, please hover your cursor over the many red and orange colored icons, shaped like keys, stars, and text bubbles.

1) I feel the authors need to very clearly acknowledge the essence of their study. The way I read the paper, the authors created a diagnostic tool, or procedure, to flag wetlands likely to be most vulnerable in the future. It bridges quantitative and qualitative research, applied in a high profile setting. Most importantly, it helps solve the problem of scale in global assessments. This product satisfies the stated goals (p. 4, line 29 through p. 5, line 7). This is a product worthy of praise and significant to both the science and management communities.

Unfortunately, the study is not presented in this light. This sets the stage for the real problem with the paper, that the authors do not consistently or explicitly discuss the limitations of the study, resulting in chronic problems with overstating conclusions - addressed in my second major comment.

To remedy the situation, I recommend:

1. Reframe the study as the development of a screening tool
2. In the Discussion/Conclusions section, refrain from restating what the results are, and instead focus on what the results mean. Specifically that they guide future research and/or allocation of resources for wetland protection
3. In the Discussion/Conclusions section, add a very clear explanation of the study limitations.
4. Avoid making conclusions about the wetlands themselves - other than to say they are likely to be vulnerable or not. Or that they are vulnerable under future simulated conditions. Perhaps discuss why hotspots exist. There absolutely is no basis to prescribe specific management action other than where to direct conservation resources and future investigation.

*Currently the focus of the paper is on the wetland analysis. We understand that the approach (i.e. the development of a screening tool) need to be more highlighted and in the focus of the paper. In accordance with the comments from the other reviewers, the discussion and conclusions section will be revised as follows: (i) The aims and main results of the study will not be repeated but limitations of the quantitative-qualitative approach will be more explicitly discussed. (ii) We will describe what the results mean for water management, river ecosystems and future research. (iii) We also agree that from the global perspective no specific management actions can be provided for single wetlands. (iv) To avoid overstating conclusions, we will make more generic statements on hotspots (not specific wetlands) and potential management options depending upon the nature of the threat as suggested by Reviewer #1.*

2) I feel the authors overstate their conclusions. However, this appears to be a symptom of a structural problem with the paper. With regard to the Methods section, the authors made many decisions on how to go about their study. This is inherent in any study and particularly messy when trying to scale up to a global assessment. Unfortunately, very little is said about the rationale for their decisions. I'm not saying I think the authors made poor decisions; I'm merely saying they should share their rationale. For example, they chose WaterGAP3 as a model. But they never state why they chose that model. I would like to see that they actually thought about other options and felt WaterGAP3 was the best for some actual reason. Also, the authors chose one particular climate change projection to use in their 50-year forecast. I'm happy they discuss this scenario as one of many. But there is no rationale provided for why they chose this one. They could add sentence and I would be happy. I flag several instances of this in my embedded comments.

More importantly, I would like to see the Discussion section evaluate the consequence of these decisions, if relevant. For example, I wonder how sensitive the results are to using that one specific climate projection. Or not considering present allocations of eFlows. This helps delineate the limitations of the study and helps guide where future study should focus.

Once the authors have thought through and delineated the limitations of the study in the Discussion section, they will be far less likely to overstate the conclusions.

*Thank you for pointing this out. In the revised version of the manuscript we will include the missing explanations:*

- *Why we chose WaterGAP3? WaterGAP3 is an integrated global modelling framework to assess impacts of global change on renewable freshwater resources. The model has been developed at the Center for Environmental Systems Research and further improved during my PhD to conduct studies for identifying river ecosystems at risk. WaterGAP3 is a state of the art global water model which performs well compared to other global models (Beck et al. 2016). Of particular interest for this study is the high spatial resolution of 5 by 5 arc minutes, the temporal resolution of daily time steps used in the analysis, the global coverage, the operation of >6000 dams with optimisation schemes for different dam types, and water withdrawals and consumption of 5 sectors (domestic, manufacturing, thermal electricity production, irrigation and livestock).*
- *Why we chose the RCP6.0 emission scenario? Current $CO_2$ emissions are close to the upper end of the scenario range. RCP6.0 is a medium-high emission scenario with a*

*global mean temperature increase of 2.2°C until the end of the century (compared to 1986-2005). The differences between the emission scenarios (as represented by the radiative forcing) are smaller than between scenarios based on different GCMs until 2050. Therefore, we considered climate forcing of 5 Global Circulation Models (GCMs) in order to address the uncertainty of projected impacts.*

- *Why current eFlow provisions are not included? Today, no global database exists that describes dam management strategies, operation rules or applied eFlow provisions of large dams. Our study benefits from the qualitative assessment where we collected legal eFlow provisions which we combined with our quantitative model outcomes. However, this data collection does not guarantee that eFlows are actually enforced and established in practice.*

*Overall, we agree that it is important to provide the rationale for our decisions more clearly. The consequences will be discussed under consideration of the limitations (model, approach) in the Discussion and Conclusions section.*

3) I feel the manuscript could better "funnel out" in the Conclusion section. This is a chance for the authors to wave their flag and tell me why their work is important. Scientifically, I'm impressed that they combined qualitative and quantitative methods to navigate issues of scale in a global assessment. They should feel free to state this as an academic contribution. The implications of this study could be far reaching. Guiding resources to protect Ramsar wetlands is a big deal. And sure, this study has its limitations. But the authors created a template that could have more broad applications. This study focused on Ramsar wetlands. But maybe the next one could be a true global wetland assessment. Maybe this template could be tweaked and applied to settings such as coral reefs, forest production, or water supply. I wish the authors would express a vision for how this study advances us in the big picture.

*In the revised manuscript we will make clear that the Ramsar sites have been chosen as an example for our screening tool and point out that many other applications of our quantitative-qualitative approach are possible. For example, the bankfull flow approach enables the assessment of all larger riparian wetlands worldwide, flood risk, and flood related processes such as temporary storage of river discharge in adjacent riparian wetlands. Other possible applications based on our study can be mentioned, e.g. the quantification of specific ecosystem services provided by intact riparian wetlands (e.g. forest production, water purification, fish production, flood control, etc.) and how this is likely to change in the future under climate change and further dam construction. It could also support the allocation of water resources to different water use sectors and the respective consequences. We will focus on providing the big picture which embeds our analysis. Thanks for this valuable remark!*

4) My one objection to their methods is the throwing out of results that suggested more overbank flooding will occur in 2050 than occurs now. This might be justified, but it sounds fishy to me. Of course climate change will cause flooding to increase in some places and decrease in others. Why wouldn't they include that in their assessment? At the very least, I recommend this decision be discussed and a rationale provided. Not providing other studies that have done similarly makes me suspicious this isn't a valid assumption. The Discussion section should provide some assessment of how this decision affected the results.

*Floods are one of the most damaging natural disasters to human lives and property. We wanted to be cautious in our paper by not labelling increasing floods as a "positive event" because many people are affected and even lose their belongings. As the focus is on riparian wetlands, we argue in the paper that "only reductions have been documented, because it cannot be distinguished whether an increase in flood volume benefits the wetland or generates flood damages, which, in turn, would be an incentive to build more dams for flood control". Nevertheless, for quantifying the increase in flood volume we could include thresholds (0-20%, 20-40%, and >40%) in the map and discuss the changes by describing the potential consequences.*

5) I sense the paper was drafted by a non-native English speaker. I am supportive of this and welcome different perspectives in the literature. Unfortunately, I had a difficult time understanding the content. If left unaddressed, I feel this will reduce the impact of the paper. I have some specific recommendations to move forward.

1. Adding subsections would greatly help keep the text organized.
2. Please keep paragraphs short and focused on the topic sentences. For example, the first paragraph of the introduction is almost a page long and drifts away from the topic sentence. That is too much.
3. Adding a flow chart to the methods section that schematically illustrates your 3 modeling exercises would be very effective at communicating what you did. I visualize your methods as having 3 'cuts' of modeling: "natural" conditions, "natural + modeled water management," and "climate change 2050 without water management." Even if/when readers become confused by the text, a nice flow chart will communicate your general approach well. See my embedded comments.
4. I feel adding a table to the Results section is essential and will allow you to delete at least half the text in the current Results section. I visualize one row per wetland in the study and one column each for wetland number, wetland name, vulnerability for the three conditions tested, and perhaps a comment column. The Results section text could be reserved to identify trends and hotspots rather than telling the reader verbally what wetlands were vulnerable. To be clear, any text that simply tells the reader "wetland X was vulnerable" could be deleted and replaced with more substantive information.
5. The Discussion section is largely dedicated to re-presenting results. In my own writing, I typically find this to be my #1 problem. Deleting any presentation of results in the Discussion section will free up vast amounts of text to focus instead on what the results mean to your study goals and the limitations of your results.
6. Similarly, conclusions are also presented in the Results section and this text should also be removed. I flag these instances in my notes on the manuscript PDF, included below within this same document.
7. Other issues exist, notably sentence structure. My sense is putting the paper through an editorial review would be the most expedient solution. There is a lot that a good editorial reviewer can add that I simply cannot.

*The methodology section contains a lot of information and the entire (modelling) approach is quite complex. We totally agree that subsections will help to better structure and improve the*

*understanding and readability of the paper. A flow chart at the beginning of the methodology is another great idea that we will adopt.*

*In line with comments of other reviewers, the discussion and conclusions section will be revised. In the new discussion section the aims of the study will not be repeated and limitations of the model and the approach applied for the assessment will be explicitly discussed. In this context we will put emphasize on interpreting our results.*

*A table with results for each wetland and condition will be implemented in the supplementary material.*

*Next to the edits made by Reviewer #1 we will take care of correct spelling.*

To conclude my thoughts on the manuscript, I feel it is publishable with major corrections. The corrections, however, are largely limited to the presentation of the paper, not a fundamental problem with the methods, per se. Again, I have added ~130 comments to the manuscript PDF, included below, to further guide revision of the manuscript. Thanks again for the opportunity to review this paper.

*We are very thankful for the specific comments in the supplementary, which were summarised in the major points above. We will address each comment appropriately in the revised manuscript.*

---

## Author Comment (AC2) · 24 Nov 2016

**Response to Referee #2**

*We thank Referee #2 for the profound evaluation of the paper and the helpful comments, which will further improve this paper. We are confident that we can adequately address each of these comments. Please find below our responses describing our planned revisions (highlighted in blue and italic type).*

Through data synthesis and model interpretations of RAMSAR wetland sites across the world, this paper addresses the issue of past to expected future adverse effects on riparian wetlands from pressures such as climate change and water regulation. In particular the focus is on the available flooding volume - how it has been modified today and how it may change in the future due to these pressures. The magnitude of these changes is taken as a measure of potential ecological impacts.

The authors combine and use multiple methods (e.g. to simulate impact of flow regulation of various dam types etc), many of which have been thoroughly developed in previous work. Although results are associated with considerable uncertainties, the approach is quite reasonable and the outcome is logically synthesised and presented as maps showing e.g. the magnitude of flow alteration impact. Such global state-ofthe-art syntheses is certainly of scientific interest; I would recommend publication of the work if main shortcomings (see below) can be addressed, which is likely to require at least moderate revisions.

In summary, these shortcomings are (1) lack of clarifications regarding novel aspects of the present study, apart from the novel global synthesis perspective, (2) partial lack of information regarding past experiences of the proposed methods, (3) language issues, (4) lack of sufficient results comparison to previous studies, and (5) unfocused conclusions. Overall, this study has high potential and I hope that the detailed comments below can be useful in addressing the current concerns.

1. Presently, the focus of the introduction is on the relevance of the topic, including what is known about vital ecosystem services of floodplain wetlands, effects of dams in a more general sense, and the need for maintaining flow variability etc. This description is on the lengthy side and could probably be condensed. However, more concrete (state-of-the art) regional examples that presumably exist in the scientific literature regarding today's impacts (or expected future impacts) on floodplain wetlands are essentially missing. Such examples should be included in the introduction, such that the readers can understand what is novel about the presented result-maps, in addition to the novel global synthesis perspective. In other words: which previous indications exist in the scientific literature regarding key results, such as the result showing that the degree of overbank flow alteration due to current management is very low in Europe (essentially green in Figure 1) whereas Australia comes out as seriously altered (or other results that are the authors think is important). I would recommend the authors to go through what they consider to be the main results of their study and make sure that the introduction informs sufficiently about the current knowledge. This would provide a necessary basis for enhancing the discussion (see bullet point 4)

*Currently the introduction is on the relevance of the topic and provides the rationale for the applied indicators. It describes the situation of wetlands worldwide, the dimension of flow regime modification due to dams, water abstractions, and water transfers (rationale indicator 1), the ecological consequences, the ecological function of floods, the ecosystem services of floods, expected future impacts on flooding regimes due to climate change and new dam initiatives (rationale indicators 2 and 3), potential measures to counteract flow regime modification, and the difficulties to*

*implement such counteractive measures, which requires legal and institutional capacity to act (rationale indicator 4). The introduction ends with three research questions describing the goal of this study.*

*We agree with Reviewer #2 that the introduction is currently too long and will shorten the section by e.g. removing the paragraph on ecosystem services of wetlands and/or condensing the text by providing only the key points and referring to the literature. Actually we provide 5 regional examples (Hughes, 1988; Maheshwari et al., 1995; Barbier and Thompson, 1998; Kingsford, 2000; Nislow et al., 2002). But we agree to the suggestion to include more regional studies and make them more prominent in the text. Following the description of current knowledge, we will derive our research question(s) and highlight the novelty of our study. In the last years, different authors have assessed ecologically relevant flow regime modifications on larger-scales. In addition and complementary to the published papers, our study considers the following points which have never been applied before in their combination and in its detail to create a screening tool for assessing hydrological threats for riparian wetlands.*

1. *Environmental flow provisions that are defined as a percentage of mean discharge can be allocated in many different ways throughout the year. However, complex flow-dependant ecosystem habitats and functions are provided by specific flow characteristics. Consequently, rather than long-term average flow conditions, our approach focuses on a specific, ecologically relevant flow event.*

2. *Most large-scale environmental flow assessments focused on in-channel river flows. Riparian wetlands depend on overbank flows leading to inundation. They are (in combination with subsequent drying) the main driving force for ecological processes in riparian wetlands. Our assessment is the first that applies the flood pulse concept (Junk et al., 1989; Bayley, 1991; Tockner et al., 2000; Junk and Wantzen, 2004) on a global scale.*

3. *In order to address trade-offs between human and ecological water demands, multiple stressors on human water security and ecosystem conservation need to be considered. The applied approach is able to consider different drivers of change such as dam operation, water use as well as climate change.*

4. *Next to the flow regime modifications, the threat for riparian wetlands also depends on the society's capacity to act to the changes. In order to fill this gap, we combined quantitative with qualitative results. The implementation of counteractive measures depends especially on the legal and institutional framework in place. Therefore, we collected 6 different criteria (legal environmental flow provisions, presence of RBOs, at least one relevant treaty, and specific treaty provisions such as water allocation mechanism, conflict resolution mechanism, and flow variability management). In addition, new dam construction is likely to further modify flow regimes in the future, but currently no large-scale dataset on major dam initiatives (including planned storage capacities) is publicly available. Therefore, we collected the number of dams that are currently planned, proposed or under construction in the upstream areas to give a first indication, where future dam construction is likely to affect the inundation of specific riparian wetlands.*

5. *Our discharge simulations were done on a daily time-step. This is important as many ecological functions and habitats are facilitated by hydrological events that last only up to some days (e.g. strong precipitation events, bankfull flow, and flood formation).*

6.      *Today, river flows are considerably affected by human activities worldwide, and the speed of river ecosystem destruction and biodiversity loss is exceeding the ability of scientists to review applied water management practices and ecological consequences for each river. Therefore this study assesses flow regime modifications on a global scale. The approach is performed on a detailed river network with a spatial resolution of 5x5 arc minutes and can be applied for single reaches of larger rivers with a global coverage.*

7.      *The approach will allow new applications related to riparian wetland flooding. Examples include the quantification of specific ecosystem services provided by intact riparian wetlands (e.g. forest production, water purification, fish production, flood control, etc.) and how this is likely to change in the future. The framework could support policy makers at international level (e.g. at forums like UNEP, OECD, European Union, Convention on Wetlands of International Importance, and Convention on Biological Diversity) in balancing water allocations to humans and nature, implementing global conservation efforts, and planning of water infrastructure location and design.*

2. It is stated in the introduction (p. 2, line 26) that a new approach is needed to water resources management, which among other things should allow for sufficiently high flows for sustaining floodplain wetlands. However, in line with comments of bullet point 1 (above), this proposed novelty remains unclear to the reader. For example, haven't we gained some relevant knowledge from regulation schemes applied to the principal Colorado River in the US (Stevens et al., 2001; Stromberg et al., 2007; Cross et al., 2011)? These schemes have included controlled floods as part of the strategy to minimise adverse impacts to downstream ecosystems. Perhaps there other relevant examples.

*Thanks for this remark. The aim of this paragraph is not to claim that the "new approach" on water resource management is our idea. Rather we want to state that both flood protection for people and controlled floods for riparian wetlands are important and need to be considered in practice within the framework of integrated water resource management. We will revise this paragraph and argue with the references mentioned by the Reviewer #2.*

3. The language of the manuscript is overall good. There are some exceptions though, including the introduction. In particular, the research questions and the related text include awkward formulations (e.g., multiple sentences starting with Thereby. . ./ Therefore. . .), please check.

*We understand that the language of this manuscript can be improved. We will check grammar, formulations and word spelling. Thanks for the given examples.*

4. There is a lack of results comparison to previous studies in the discussion section, which should be addressed before publication. The now included references do mainly not relate to the results (study outcomes) and need therefore to be complemented. For instance, are the results regarding impacts on the 93 Ramsar wetlands in different world regions (p. 17, lines 3-11) consistent with previously reported results for these regions? Alternatively, do the results partly contradict or point to new and previously unnoticed aspects? (Also, the reader is not well informed about the existence or absence of similar studies, see bullet point 2 above regarding the introduction). The same questions can be asked for other key results, such as impacts of climate change and the related identified hotspots (p. 17, line 12-15), and competition of water (p. 17, lines 31-32). Overall, the discussion section is rather

general and would benefit from an extended discussion of results. The aims of the study need not to be reiterated in the beginning of the discussion section.

*The discussion and conclusions section will be revised according to the advices given by the three reviewers. With regards to the specific comments of Reviewer #2 we will substantiate the discussion part by comparing our key findings with other existing studies on a regional basis or even for specific wetlands which forms again a bracket with the introduction section. This will be done not only for the findings but also for the interpretation of the data and new insights gained by our quantitative-qualitative approach.*

5. The main conclusions of the paper are not clearly presented. Maybe a separate conclusion section could help?

*Thanks for this remark. In our revised manuscript we will put particular attention on the revision of the conclusions section. We ensure to clearly present our conclusions aiming at the novelty of this study identified in the Introduction section. Additionally we will include a sub-section on future research and the potential of our approach to be applied to similar questions related to other ecosystems at risk.*

References

Cross, W. F., Baxter, C. V., Donner, K. C., Rosi-Marshall, E. J., Kennedy, T. A., Hall, R. O., ... & Rogers, R. S. (2011). Ecosystem ecology meets adaptive management: food web response to a controlled flood on the Colorado River, Glen Canyon. Ecological Applications, 21(6), 2016-2033.

Stevens, L. E., Ayers, T. J., Bennett, J. B., Christensen, K., Kearsley, M. J., Meretsky, V. J., ... & Springer, A. E. (2001). Planned flooding and Colorado River riparian trade-offs downstream from Glen Canyon dam, Arizona. Ecological Applications, 11(3), 701-710.

Stromberg, J. C., Beauchamp, V. B., Dixon, M. D., Lite, S. J., & Paradzick, C. (2007). Importance of low-flow and high-flow characteristics to restoration of riparian vegetation along rivers in arid south-western United States. Freshwater Biology, 52(4), 651-679.

---

## Author Comment (AC3) · 25 Nov 2016

**Response to Referee #3**

*We thank Referee #3 for the profound evaluation of the paper and the helpful comments, which will further improve this paper. We are confident to adequately address each comment and our reply describing the planned revisions of the manuscript are highlighted in blue and italic type.*

The authors aim to address an important issue: Identifying Ramsar riparian wetlands that exhibit current and future variations in ecologically consequential inundation patterns as a result of human-modified flows (e.g., dams). They ask three particular research questions to best identify these wetlands. These questions focus on the impact of current water resource management on riparian wetland flows, the effect of future climate change on inundation of these wetlands, and the implications of low government and societal infrastructure and capacity to make changes to future management.

The goal and research questions the authors attempt to address are broad and could be impactful if addressed and translated well. However, a major revision is required to ensure both the quantitative work behind the research and the communication of this work is effective. Below, I provide major suggestions for the manuscript followed by some general comments.

Major Point 1: The Introduction reads somewhat like a full literature review that continues for quite some time without a direct point. It was well into the sixth paragraph that the goal and research questions were stated. I would suggest tightening up the Introduction, providing only key points throughout, and early on (perhaps at the end of the first paragraph) allude to the main point of the paper (e.g., "We aim to. . ."). Then, the authors can safely state the full objective and research questions at the end of the Introduction.

*As already stated in the reply to Reviewer #2, the Introduction section provides information on the relevance of the topic and the rationale for the applied indicators. It describes the situation of wetlands worldwide, the dimension of flow regime modification due to dams, water abstractions, and water transfers (rationale indicator 1), the ecological consequences, the ecological function of floods, the ecosystem services of floods, expected future impacts on flooding regimes due to climate change and new dam initiatives (rationale indicators 2 and 3), potential measures to counteract flow regime modification, and the difficulties to implement such counteractive measures, which requires legal and institutional capacity to act (rationale indicator 4). The introduction ends with three research questions describing the goal of this study.*

*We agree with the reviewer that the introduction should be shorten and allude the main points of the paper earlier in the text. We intent to shorten the text by e.g. removing the paragraph on ecosystem services of wetlands and/or condensing the text by addressing key points which will be underpinned with references. The revised introduction will be less like a literature review but more focused on specific examples to derive the novelty of our approach.*

Major Point 2: Something is very misleading and incorrect about discussing a "natural" flow regime in knowingly modified watersheds and aquatic systems. Also, the word "natural" is used throughout the Abstract and Introduction (ala Poff et al. 1997), and it is not until the Methodology that authors define natural flow. The authors describe natural flow for this paper as "simulated taking into

account current climate and landcover conditions, but no further anthropogenic impacts." This, by no means, would constitute a "natural" flow regime as described in past literature. I would recommend modifying terminology and the discussion throughout the paper to consider this as your "baseline" flow regime from which the analyses aims to understand current water resource management implications on the riparian wetlands and project changes of these regimes due to climate change

*We will make use of the terminology "reference flow regime" as we agree with Reviewer #3 that the use of the terminology 'natural flow regime' is misleading due to the fact that current climate change and land cover conditions are considered. The new terminology will be applied in the main body of the manuscript as well as in the Abstract.*

Major Point 3: The goals of the paper and research questions are poorly worded need more information. What, specifically, are the "riparian wetlands?" In the Abstract, the authors suggest they look at 93 Ramsar sites. Are the "riparian wetlands" the "93 Ramsar riparian wetlands?" For Research Question 1, why are 6025 dams selected? Are these dams specifically located upstream of Ramsar riparian wetlands? What are the "different water use sectors"? Also, delete "Thereby" at the beginning of the second sentence. For Research Question 2: "Inundation" cannot be "impaired" because "inundation" does not necessarily denote a positive quality. The authors could replace "impaired" with "exacerbated or diminished" or "modified." Also, delete "Therefore" at the beginning of the second sentence. Research Question 3 is stated in a grammatically incorrect way, so it took a few re-reads to understand it. Move "could" after the word "sites." Also, what is a "low capacity to act?" This is definitely not clear.

*Thanks for these remarks. We understand that the research questions need to be rephrased. The proposed corrections will be implemented and the objectives will be described in more detail.*

Major Point 4, WaterGAP3 runs: Streamflow, for what the authors term "daily natural flow regimes" (1981-2010), is simulated with 2004 land cover. Using 2004 land cover is okay; however, going back to the use of the word "natural". . .how can this be considered natural flow when the landscape for each area is likely highly modified and streamflow is a reflection of these anthropogenic activities? Also, there is no mention of calibration and verification of the model, which admittedly would be difficult a global scale. Therefore, is the entire paper a thought experiment using an uncalibrated global model to help explore hypotheses? It would be okay if so, though this framework should be characterized as such throughout the paper. Also, the results (maps, in particular) should emphasize the paper's overarching approach (i.e., the thought experiment – a "screening tool" is mentioned in the Discussion, hypotheses testing, and/or a conceptual model). If calibration and verification did occur at some stage and is not referenced, again, measured streamflow would reflect the managed conditions, not some unattainable "natural" or "near natural" condition. The model scenarios are therefore a bit confusing and need some rethinking, definitely in the presentation of what they are but potentially in which ones should be used. For example, consideration should focus on whether only the managed scenario and future climate/management conditions should be used since the true "natural flow regimes" aren't captured.

Also, the authors talk about the database of dams that are used, but how does that relate back to the Ramsar wetlands? Are these dams all upstream of Ramsar wetlands? As I read on, it became a bit clearer that this is simply a global database, and Ramsar wetland areas within the global domain are analyzed. However, this information (spatial domain and selection of dams) needs to be clearer up front.

The authors likely have all the information mentioned in this Major Point. There is simply a need for better and clearer communication regarding these bits of information. As a result, the Methodology section seems quite disjointed and leaves the reader guessing at how the authors conducted the analyses.

*We want to thank Reviewer #3 for these valuable comments. First, the comment on terminology has already been addressed under 'Major Point 2'. Second, the WaterGAP3 model is calibrated and validated which is described in the manuscript at p.7 line 32 to p.8 line 3. For the verification of the model, we refer to p.8 line 15 and the reference Schneider et al. (2011a) given in the text, which contains details of the model performance with regards to bankfull flow events. However, given this comment we think there is the need to improve the structure of the Methods section and to better explain the model used in this study. We will address this comment by including sub-headings and an extra paragraph on model calibration and validation to increase clarity, transparency and understanding of the paper. Furthermore we will add another reference on WaterGAP3's ability to represent maximum flow magnitudes (Schneider 2015).*

*The entire paper is far from being a thought experiment, although any model experiment could be understood as a 'thought experiment'. To be more precise, in this paper we used a model to estimate the hydrological threats for riparian wetlands today and in the future (2050s) under the conditions of a selected scenario. Hydrological models are useful tools to mirror the reality, i.e. river discharge, in an abstract manner. The higher the agreement of simulated and observed data records the better the model performance. In the calibration process, WaterGAP3 model simulations take into consideration human impacts in terms of managed reservoirs and dams, water abstractions and return flows from 5 different sectors, urban water transfers and land use conditions. In general, the model is calibrated against an observed discharge record by adjusting one free parameter (runoff coefficient) and validated to an independent period of the same discharge record. Based on the calibrated model the 'reference flow regime' is represented by a model simulation driven solely by the meteorological forcing of the respective time period, i.e. 1981 to 2010. Human interventions in form of managed dams and reservoirs, water abstraction, return flows, urban water transfers are omitted in this model simulation. This model simulation can be compared to the respective time period including the human interventions to evaluate their impacts. However, we want to mention that this approach as well as the terminology 'natural flow' is commonly used in the community of global hydrological modellers. Recognising the misunderstanding of the information given in the text we will improve the manuscript by providing model-specific information and references to the model calibration and validation and conduct a thorough revision of the entire text. This will also include the description of the scenario selected to identify future hotspots under varying climate and socio-economic conditions.*

*The dam database (GRANd) used by the model is initially independent of the Ramsar wetlands. GRANd contains the information on the location, storage capacity and main purpose of the largest dams of the world (about XX% of the total dam storage) which are not necessarily located upstream of Ramsar wetlands. We decided to focus on the Ramsar wetlands because of their importance and description found in scientific literature. We will communicate this in a clearer way in the text.*

Major Point 5, Discussion and Conclusions: Be careful here. Because this is thought experiment using a global model (again, unless calibration/verification happened but wasn't mentioned), your conclusions need to be balanced with a statement of the conceptual aims of the paper and

associated limitations/assumptions. The quantitative analyses isn't incredibly quantitative, and I wince a bit with the use of numbers like "8% are significantly impaired" and flood volume is likely to be decreased at 41% of the sites. . ." when those are all relative numbers with no basis in reality. Please mention up front in the conclusions or make a separate section of the limitations and assumptions with regard to what the analyses can actually provide.

*We will improve the discussion and conclusions section taking into account a cautious valuation of our model outcomes and resulting conclusions. An improvement of the discussion part will be achieved by comparing our key findings with other studies preferably related to the selected Ramsar sites. This includes the studies already mentioned in the introduction section but will be extended through systematic literature search.*

Major Point 6: In general, the English is okay as written. However, it's important that someone extremely proficient in English re-review this paper for odd placement of verbs, adjectives, modifiers, etc., and poor word selection. One small example, on Page 5, Line 14 "For Europe, a higher number of sites "were gained" as the European wetland geodatabase. . .". This should be "were selected" or "were chosen". There are many instances like this throughout the paper, and I do not list them all below.

*We agree that the language of this manuscript needs to be improved. We will check again grammar, wording and spelling. Thanks for the given examples.*

Specific Comments

Page 1, Line 9 – Recommend changing all references of "mankind" to "humankind" and "man-made" to "constructed"

*Thanks, we will modify all terms, respectively.*

Page 1, Line 9 – These eco services are provided not only via the regular patterns of inundation but also regular patterns of drying – so actually, it's the *variability* inundation patterns that is important.

*Thanks, we will modify this.*

Page 1, Line 26 – Need to review and add Dixon et al (2016) as well. Dixon, MJR, Loh J, Davidson NC et al. 2016. Tracking global change in ecosystem area: The Wetland Extent Trends index. Biol. Conserv. 193: 27-35.

*Thanks, we will include this reference.*

Page 2, Lines 18-19 – Is this true for all "larger cities?" What spatial scale is this referring to? Are these global or regional estimates? If regional, what regions?

*Thanks, we will specify this and add a reference (Mc Donald et al. 2014).*

Page 2, Line 25 – Again, what are "natural sites?"

*Thanks, we will provide a definition in the revised manuscript.*

Page 2, Line 30 – Not all floodplains are wetlands, which is how this sentence reads. Please correct.

*Thanks, we will correct this.*

Page 2, Line 32 – What ecological processes are initiated? Some of these processes may be initiated by drying not wetting.

*Thanks, we will include examples..*

Page 3, Line 2 – What is engendering what? This clause doesn't make sense.

*Thanks. This sentence will be rephrased to make clear that a periodically flooding and drying "engenders one of the most dynamic, diverse and productive systems in the world".*

Page 3, Line 4 – That's a very broad statement, that all floodplain wetlands contain more species than any other landscape unit. Need more specifics here because it's likely not what the authors intended to say.

*Thanks. We will be more specific in the revised manuscript.*

Page 3, first paragraph – The Roman numerals are not needed when providing full sentences after them. Suggest removing all Roman numerals here.

*Thanks, we will remove the Roman numerals.* (*However, this paragraph will most likely be removed in order to shorten the introduction.*)

Page 3, Line 24 - What are "fellow riparians?" Please be more specific.

*Thanks, we will replace it by "upstream/downstream water users"*

Page 3, Line 25 – What projections? Please be more specific.

*Thanks, we will specify this.*

Page 4, Lines 12-16 – Break up this sentence into two or more sentences.

*Thanks, we agree.*

Page 5, Lines 19-20 – These sentences can be deleted and are unnecessary.

*Thanks, we agree.*

Page 5, Line 23 – "percent change in flood volume": from what period to what period? Please provide time frame.

*Thanks, we will specify the time period.*

Page 5, Line 28 – It is not clear at this point what "sufficient capacity to act" means. Suggest modifying this or adding some clarification here to lead the reader to the more specific methods discussion.

*Thanks, we will add an explanation at the beginning of the respective section.*

Page 5, Line 3 – The simulation of daily natural flow regimes would still be an expression of a modified landscape, so how are these natural?

*Thanks, we will change the terminology.*

Page 5, Lin 9 – Need clarification of what type of "daily river discharge" is being simulated here – "natural" or "managed"? (After reading on, it becomes obvious it's "natural" but that needs to be mentioned straight away.)

*Thanks, we will add a sentence for clarification.*

Page 5, Line 20 – Switched to "near-natural" from "natural" in this sentence. Please be consistent.

*Thanks, we will change the terminology.*

Page 7, Lines 6-9 – Need to be clear here why the simulation includes these specific 6025 dams. Why were they chosen? Intuition would tell me they are all upstream of Ramsar sites, but further reading seems to suggest that they are simply part of the global database. These questions regarding methods also suggest that clear summary statements of what the quantitative analyses is up front in the Methodology should be added – meaning state your steps: exact simulations, the spatial scale, how dams were selected, how the Ramsar sites were overlain on the global map, etc. Then, details can be added after this summary.

*See reply to Major Point 5*

Page 9, line 14 – What selected sites? The Ramsar wetlands? Again, details are needed here.

*Thanks, we will revise the text.*

Page 9, Lines 14-15 – This sentence is a bit wonky and needs to be reworded.

*Thanks, we will reword the sentence.*

Page 9, Line 18 – How were the cutoff thresholds for Table 2 selected?

*Thanks, we will include the missing information.*

Page 9, Line 20 – Again, clarify what the "low capacity to act" is.

*Thanks, we will specify this.*

Page 9, Line 27 – Define blue water

*Thanks for this remark. The term "blue water footprint" is used in the cited literature and probably unnecessary jargon. We will rephrase this and describe that the scarcity threshold is reached when 20% of the streamflow is depleted.*

Page 9, Line 30 – Again, how were the Table 3 thresholds derived?

*Thanks, we will include the missing information.*

Page 10, Lines 25-27 - Cut these sentences. Too much introduction here.

*Thanks, we agree and will remove the mentioned sentences.*

Page 11, Line 4 – These wetlands are "moderately impacted" – as far as the map seems to read.

*Thanks, we will be more specific about the location to avoid misunderstandings.*

Page 11, Line 17 – N=2, though, correct? So this is only discussing two wetlands, right?

*Thanks, we will rephrase the sentence to avoid generalisation.*

Page 12, Line 28 – Is this the ensemble median for the GCMs as input to the WaterGAP3 model or the ensemble average of the output of the WaterGAP3 model?

*Thanks for this remark. We will be more precise and rephrase the sentence. It's the ensemble average of the output of the WaterGAP3 model.*

Page 13, Liens 16-17 – Now that is a very interesting finding!

*Thanks.*

Table 1, change "not/slightly" to "none/slightly" – same with the figures: "not/slightly" does not make sense.

*Thanks, we will change the text as suggested.*

Table 2, delete "the number of" in the caption.

*Thanks, will be deleted.*

Table 4, define "formal institutional capacity" in the caption to make the table stand alone.

*Thanks, we will include the missing information.*

The final edits for the paper are included in the Major Points listed previously.

---

## Author Response (AR1)

*Kassel, 31 January 2017*

*Dear Nandita Basu,*

*thank you for the positive evaluation of our manuscript. We have now incorporated all points raised by the reviewers. In particular, we aimed at improving the presentation quality of our work. Instead of the analysis, the manuscript now concentrates on our approach as a screening tool to flag particularly vulnerable wetlands. The Chapters 'Discussion' and 'Conclusions' are more focused now describing the meaning of our results as well as limitations and potential applications of our approach. The Introduction was shortened and includes now an extra paragraph on global eFlow assessments of the past. Additionally, we highlight now more the novelty of our approach. The 'Methodology' Chapter is better structured. Here, we included subheadings and a flow chart displaying the single steps of our approach. Under 'Results' we focus on the hotspots now (rather than on single wetlands) and included more comparisons to the literature.*

*We highly appreciate the scientific discussion that took place and believe that the manuscript has improved significantly on the basis of the valuable and deliberate reviewer comments. Following the point-by-point responses to the three reviewers, this document also contains a marked-up manuscript version with all changes highlighted. In the supplement we added Appendix C where we provide results for each single climate projection and thus show the uncertainty between the five chosen climate models.*

*We have uploaded the revised manuscript, abstract and supplement as separate files.*

*Kind regards,*
*Christof Schneider (on behalf of all authors)*

**Response to Referee #1**

*Dear Referee #1,*
*we are very thankful for the profound and helpful evaluation of the paper and the detailed comments in the PDF version, which certainly improved this paper. We understood that we needed to work especially on the presentation of our results and included your suggestions. Additionally rationales are provided for our methods. Please find below our point-by-point responses describing our revisions in detail (highlighted in blue and italic type).*

Hello Ms. Töpfer.

Thank you for the opportunity to provide an internal review of the paper Hydrological threats for riparian wetlands of international importance – a global quantitative and qualitative analysis. The authors have created an innovative and useful screening tool to flag particularly vulnerable wetlands. I am particularly impressed by the integration of quantitative and qualitative methods to achieve their goals. This will be a strong contribution to the literature and useful to organizations seeking to target wetland conservation funds. In addition, the same schematic approach can be applied far beyond wetlands, further enhancing the potential impact of this work. Below, I discuss the major suggestions I have for the manuscript. Finally, I have added comments well over 100 comments directly to the PDF version of the paper and figures, included below within this same document. To read my embedded comments, please hover your cursor over the many red and orange colored icons, shaped like keys, stars, and text bubbles.

1) I feel the authors need to very clearly acknowledge the essence of their study. The way I read the paper, the authors created a diagnostic tool, or procedure, to flag wetlands likely to be most vulnerable in the future. It bridges quantitative and qualitative research, applied in a high profile setting. Most importantly, it helps solve the problem of scale in global assessments. This product satisfies the stated goals (p. 4, line 29 through p. 5, line 7). This is a product worthy of praise and significant to both the science and management communities.

   Unfortunately, the study is not presented in this light. This sets the stage for the real problem with the paper, that the authors do not consistently or explicitly discuss the limitations of the study, resulting in chronic problems with overstating conclusions - addressed in my second major comment.

   To remedy the situation, I recommend:

   1. Reframe the study as the development of a screening tool
   2. In the Discussion/Conclusions section, refrain from restating what the results are, and instead focus on what the results mean. Specifically that they guide future research and/or allocation of resources for wetland protection
   3. In the Discussion/Conclusions section, add a very clear explanation of the study limitations.
   4. Avoid making conclusions about the wetlands themselves - other than to say they are likely to be vulnerable or not. Or that they are vulnerable under future simulated conditions. Perhaps discuss why hotspots exist. There absolutely is no basis to prescribe specific management action other than where to direct conservation resources and future investigation.

*Before the review process the focus of the paper was on the wetland analysis. We understood that the approach (i.e. the development of a screening tool) need to be more highlighted and in the focus of this paper. Therefore, we reframed the study as suggested. In accordance with the comments, the paper was revised as followed:*

*We divided the 'Discussion and clonclusions' chapter into two separate chapters. Under 'Discussion', we explicitly discuss now the limitations of our quantitative-qualitative approach and the consequences of our decisions. Afterwards we describe what the results mean for water management depending on the nature of the threat. Aims and main results of the study are not repeated anymore. Furthermore, we removed all conclusions for single wetlands to avoid overstating our results. We agree that from the global perspective no specific management actions can be provided for single wetlands and this is also not the aim of our screening tool.*

*Under 'Conclusions and outlook', we conclude and address future research and potential applications of our approach.*

2) I feel the authors overstate their conclusions. However, this appears to be a symptom of a structural problem with the paper. With regard to the Methods section, the authors made many decisions on how to go about their study. This is inherent in any study and particularly messy when trying to scale up to a global assessment. Unfortunately, very little is said about the rationale for their decisions. I'm not saying I think the authors made poor decisions; I'm merely saying they should share their rationale. For example, they chose WaterGAP3 as a model. But they never state why they chose that model. I would like to see that they actually thought about other options and felt WaterGAP3 was the best for some actual reason. Also, the authors chose one particular climate change projection to use in their 50-year forecast. I'm happy they discuss this scenario as one of many. But there is no rationale provided for why they chose this one. They could add sentence and I would be happy. I flag several instances of this in my embedded comments.
More importantly, I would like to see the Discussion section evaluate the consequence of these decisions, if relevant. For example, I wonder how sensitive the results are to using that one specific climate projection. Or not considering present allocations of eFlows. This helps delineate the limitations of the study and helps guide where future study should focus. Once the authors have thought through and delineated the limitations of the study in the Discussion section, they will be far less likely to overstate the conclusions.

*Thank you for pointing this out. In the revised version of the manuscript we included the missing explanations:*

- *Why we chose WaterGAP3? WaterGAP3 is an integrated global modelling framework to assess impacts of global change on renewable freshwater resources. The model has been developed at the Center for Environmental Systems Research and further improved recently to represent specific flow events (Verzano and Menzel, 2009; Verzano et al., 2012) and during my PhD to conduct studies for identifying river ecosystems at risk (Schneider et al., 2013). WaterGAP3 is a state-of-the-art global water model which performs well compared to other global models (see Beck et al. 2016, Eisner et al. 2017). The model's ability to represent specific flow events has*

*been proven for different maximum flow magnitudes (Schneider et al., 2011a; Schneider, 2015; Eisner, 2016). Of particular interest for this study is the high spatial resolution of 5 by 5 arc minutes, the temporal resolution of daily time steps used in the analysis, the global coverage, the operation of >6000 dams with optimisation schemes for different dam types, and water withdrawals and consumption of 5 sectors (domestic, manufacturing, thermal electricity production, irrigation and livestock). Furthermore, we discuss now potential improvements of WaterGAP3 in regard to this analysis in the discussion chapter.*

- *Why we chose the RCP6.0 emission scenario? Current $CO_2$ emissions are close to the upper end of the scenario range. RCP6.0 is a medium-high emission scenario with a global mean temperature increase of 2.2°C until the end of the century (compared to 1986-2005). Within the future time frame, the differences between the emission scenarios (as represented by the radiative forcing) are smaller than between scenarios based on different GCMs until 2050. Therefore, we considered climate forcing of 5 Global Circulation Models (GCMs) in order to address the uncertainty of projected impacts.*

- *Why current eFlow provisions are not included? Today, no global database exists that describes dam management strategies, operation rules or applied eFlow provisions of large dams. Our study benefits from the qualitative assessment where we collected legal eFlow provisions which we combined with our quantitative model outcomes. However, this data collection does not guarantee that eFlows are actually established in practice, enforced or adequate. As most of this information on legal eFlow provisions was available in qualitative terms we introduce a simple yes-no query to our capacity to act indicator. In particular no quantitative information on eFlow provisions was found for the management of dams.*

*Overall, we agree that it is important to provide the rationale for our decisions more clearly. The consequences are also discussed under consideration of the limitations (i.e. consideration of 5 GCMs but only one emission scenario, focus on large dams, no consideration of eFlow provisions in our dam operation scheme) in the Discussion and Conclusions section.*

3) I feel the manuscript could better "funnel out" in the Conclusion section. This is a chance for the authors to wave their flag and tell me why their work is important. Scientifically, I'm impressed that they combined qualitative and quantitative methods to navigate issues of scale in a global assessment. They should feel free to state this as an academic contribution. The implications of this study could be far reaching. Guiding resources to protect Ramsar wetlands is a big deal. And sure, this study has its limitations. But the authors created a template that could have more broad applications. This study focused on Ramsar wetlands. But maybe the next one could be a true global wetland assessment. Maybe this template could be tweaked and applied to settings such as coral reefs, forest production, or water supply. I wish the authors would express a vision for how this study advances us in the big picture.

*In the revised manuscript we make clearer now that the Ramsar sites have been chosen as an example for our screening tool. In the conclusions and outlook chapter, we point out that many other applications of our quantitative-qualitative approach are possible. (i) A comprehensive assessment of all larger riparian wetlands worldwide. (ii) The quantification of*

*specific ecosystem services provided by riparian wetlands (e.g. forest production, water purification, fish production, flood control, etc.) and how this is likely to change in the future under climate change and further dam construction. (iii) A comprehensive scenario assessment considering different drivers of global change on renewable freshwater resources by allocating water resources to different water use sectors and evaluating the respective consequences under different management targets. (iv) Support of policy makers at international level in implementing global conservation efforts, targeting wetland conservation funds, planning of water infrastructure location and design, and balancing water allocations to humans and nature. We provide now the big picture which embeds our analysis. Thanks for this valuable remark!*

4) My one objection to their methods is the throwing out of results that suggested more overbank flooding will occur in 2050 than occurs now. This might be justified, but it sounds fishy to me. Of course climate change will cause flooding to increase in some places and decrease in others. Why wouldn't they include that in their assessment? At the very least, I recommend this decision be discussed and a rationale provided. Not providing other studies that have done similarly makes me suspicious this isn't a valid assumption. The Discussion section should provide some assessment of how this decision affected the results.

*Floods are one of the most damaging natural disasters to human lives and property. We wanted to be cautious in our paper by not labelling increasing floods as a "positive event" because many people are affected and even lose their belongings. As the focus is on riparian wetlands, we argue in the paper that "only reductions have been documented, because it cannot be distinguished whether an increase in flood volume benefits the wetland or generates flood damages, which, in turn, would be an incentive to build more dams for flood control". Nevertheless, we understood the point raised and quantify now the increase in flood volume by two thresholds distinguishing a low (0-30%) and high (>30%) increase. We felt that more classes would have made the map unclear and difficult to read. However, it enabled us to identify hotspots where flood volumes highly increase in our simulations. The outcomes are provided in the results chapter. Furthermore, we discuss the increasing flood volumes in the discussion chapter by describing the potential consequences for water resource management. We agree that providing these results is an improvement of the paper.*

5) I sense the paper was drafted by a non-native English speaker. I am supportive of this and welcome different perspectives in the literature. Unfortunately, I had a difficult time understanding the content. If left unaddressed, I feel this will reduce the impact of the paper. I have some specific recommendations to move forward.

   1. Adding subsections would greatly help keep the text organized.

      *The methodology section contains a lot of information and the entire (modelling) approach is quite complex. As suggested, we included subsections in the Chapters 'Methodology' and 'Results' to better structure the paper and improve the understanding/readability. The subheadings correspond to the flow chart.*

   2. Please keep paragraphs short and focused on the topic sentences. For example, the first paragraph of the introduction is almost a page long and drifts away from the topic sentence. That is too much.

      *We revised all paragraphs to make them shorter and more focused on the topic sentence.*

3. Adding a flow chart to the methods section that schematically illustrates your 3 modeling exercises would be very effective at communicating what you did. I visualize your methods as having 3 'cuts' of modeling: "natural" conditions, "natural + modeled water management," and "climate change 2050 without water management." Even if/when readers become confused by the text, a nice flow chart will communicate your general approach well. See my embedded comments.

   *A flow chart at the beginning of the methodology section was another great idea that we adopted. The flow chart corresponds to the new subheadings.*

4. I feel adding a table to the Results section is essential and will allow you to delete at least half the text in the current Results section. I visualize one row per wetland in the study and one column each for wetland number, wetland name, vulnerability for the three conditions tested, and perhaps a comment column. The Results section text could be reserved to identify trends and hotspots rather than telling the reader verbally what wetlands were vulnerable. To be clear, any text that simply tells the reader "wetland X was vulnerable" could be deleted and replaced with more substantive information.

   *A table with detailed results for each wetland and threat was implemented in the supplementary material. The 'Results' Chapter was shortened and focuses now on hotspots and trends rather than single wetlands.*

5. The Discussion section is largely dedicated to re-presenting results. In my own writing, I typically find this to be my #1 problem. Deleting any presentation of results in the Discussion section will free up vast amounts of text to focus instead on what the results mean to your study goals and the limitations of your results.

   *The discussion and conclusions section was revised. In the new discussion section the aims of the study are not repeated anymore and limitations of the approach are explicitly discussed. In this context we put emphasize on interpreting our results. (see also major point 1)*

6. Similarly, conclusions are also presented in the Results section and this text should also be removed. I flag these instances in my notes on the manuscript PDF, included below within this same document.

   *All conclusions were removed in this section.*

7. Other issues exist, notably sentence structure. My sense is putting the paper through an editorial review would be the most expedient solution. There is a lot that a good editorial reviewer can add that I simply cannot.

   *We checked again grammar, sentence structure and wording and hope we could improve it. Thanks for the given examples!*

To conclude my thoughts on the manuscript, I feel it is publishable with major corrections. The corrections, however, are largely limited to the presentation of the paper, not a fundamental problem with the methods, per se. Again, I have added ~130 comments to the manuscript PDF, included below, to further guide revision of the manuscript. Thanks again for the opportunity to review this paper.

*We are very thankful for the specific comments in the supplementary, which were summarised in the major points above. We addressed each comment appropriately in the revised manuscript. Please find answers below that are not addressed in the major points above.*

- *How did we choose riparian wetlands? The Ramsar Classification System describes different wetland types, but does not categorize riparian wetlands. However, riparian wetlands were selected from the Ramsar list on the basis of information provided by the Ramsar information sheets (RSIS, no date). For each wetland we read the information sheet and a wetland was selected when the information sheet indicated that a wetland depends (at least partly) on flooding by adjacent rivers. For Europe, a higher number of sites could be selected as the European wetland geodatabase (Okruszko et al., 2011) clearly defines wetland type and main source of water for each Ramsar wetland. Here we took into account all wetlands with rivers as a main source of water.*

- *Why we focus on large dams? In order to assess flow alterations due to dam operation, the number of dams implemented in the model has been further increased for this paper. We operate now >6000 dams which is state-of-the-art in comparison to other global models (see supporting information of Haddeland et al., 2013). In order to have a clear cut, we decided to consider all large dams + smaller dams with a storage volume >0.5 km$^3$. In the discussion chapter we discuss the consequences of this decision.*

- *A basin country unit (BCU) is the portion of a country within a river basin shared by two or more countries. The description of our approach was not clear at this point and so we reformulated our proceeding. Thanks for pointing this out.*

- *Why did we not model climate change together with water management? We agree that future water management plays a crucial role, especially in regard to how we respond to climate change. We decided to focus only on climate change because a high number of new dams is expected in the future and important information on dam type and storage capacity was not available for us. We are still working on collecting this information and further improvements of the screening tool will address the implementation of future dams in the model. Although model outcomes of tier 2 will not reflect future conditions because of not taking into account future water management, this model experiment supports identifying the solely effect of climate change on riparian wetland inundation.*

**Response to Referee #2**

*Dear Referee #2,*
*we are very thankful for the profound evaluation of the paper and the helpful comments, which further improved this paper. Please find below our point-by-point responses describing our revisions (highlighted in blue and italic type).*

Through data synthesis and model interpretations of RAMSAR wetland sites across the world, this paper addresses the issue of past to expected future adverse effects on riparian wetlands from pressures such as climate change and water regulation. In particular the focus is on the available flooding volume - how it has been modified today and how it may change in the future due to these pressures. The magnitude of these changes is taken as a measure of potential ecological impacts.

The authors combine and use multiple methods (e.g. to simulate impact of flow regulation of various dam types etc), many of which have been thoroughly developed in previous work. Although results are associated with considerable uncertainties, the approach is quite reasonable and the outcome is logically synthesised and presented as maps showing e.g. the magnitude of flow alteration impact. Such global state-ofthe-art syntheses is certainly of scientific interest; I would recommend publication of the work if main shortcomings (see below) can be addressed, which is likely to require at least moderate revisions.

In summary, these shortcomings are (1) lack of clarifications regarding novel aspects of the present study, apart from the novel global synthesis perspective, (2) partial lack of information regarding past experiences of the proposed methods, (3) language issues, (4) lack of sufficient results comparison to previous studies, and (5) unfocused conclusions. Overall, this study has high potential and I hope that the detailed comments below can be useful in addressing the current concerns.

1. Presently, the focus of the introduction is on the relevance of the topic, including what is known about vital ecosystem services of floodplain wetlands, effects of dams in a more general sense, and the need for maintaining flow variability etc. This description is on the lengthy side and could probably be condensed. However, more concrete (state-of-the art) regional examples that presumably exist in the scientific literature regarding today's impacts (or expected future impacts) on floodplain wetlands are essentially missing. Such examples should be included in the introduction, such that the readers can understand what is novel about the presented result-maps, in addition to the novel global synthesis perspective. In other words: which previous indications exist in the scientific literature regarding key results, such as the result showing that the degree of overbank flow alteration due to current management is very low in Europe (essentially green in Figure 1) whereas Australia comes out as seriously altered (or other results that are the authors think is important). I would recommend the authors to go through what they consider to be the main results of their study and make sure that the introduction informs sufficiently about the current knowledge. This would provide a necessary basis for enhancing the discussion (see bullet point 4)

*We agree that the introduction was too long and information about current knowledge needed to be improved. Accordingly we condensed the text by often providing only key points and referring to the literature. The more detailed descriptions of the flood pulse concept and the paragraph on ecosystem services of wetlands were removed.*

*We provide 6 regional examples (Hughes, 1988; Maheshwari et al., 1995; Barbier and Thompson, 1998; Kingsford, 2000; Nislow et al., 2002; Middelkoop et al., 2015) in the introduction to show that floodplain wetlands have been downsized and transformed into terrestrial ecosystems due to reduced flooding caused by water resource management. More comparisons to the outcomes of these studies as well as other studies (Georgiyevsky et al., 1995; 1996; 1997; Khublaryan, 2000; Tockner and Stanford, 2002; Uluocha and Okeke, 2004; UNEP, 2008; Schneider et al., 2011b; 2013; Dankers et al., 2013, Zarfl et al., 2014; Grill et al., 2015) are made now in the 'Results' chapter.*

*Following the description of the current situation and providing the rationale for our indicators, we added an additional paragraph to the introduction. This paragraph mentions valuable studies of the past (Smakhtin and Eriyagama, 2008; Döll et al., 2009; Döll and Zhang, 2010; Schneider et al., 2013; Laize et al. 2014; Pastor et al., 2014; Grill et al, 2015) addressing flow regime alterations on larger-scales and highlights the novelty of our study. In the last years, different authors have assessed ecologically relevant flow regime modifications on larger-scales. In addition and complementary to the published papers, our study considers the following points which have never been applied before in their combination and in its detail to create a screening tool for assessing hydrological threats for riparian wetlands:*

1. *Environmental flow provisions that are defined as a percentage of mean discharge can be allocated in many different ways throughout the year. However, complex flow-dependant ecosystem habitats and functions are provided by specific flow characteristics. Consequently, rather than long-term average flow conditions, our approach focuses on a specific, ecologically relevant flow event.*

2. *Most large-scale environmental flow assessments focused on in-channel river flows. Riparian wetlands depend on overbank flows leading to inundation. They are (in combination with subsequent drying) the main driving force for ecological processes in riparian wetlands. Our assessment is the first that applies the flood pulse concept (Junk et al., 1989; Bayley, 1991; Tockner et al., 2000; Junk and Wantzen, 2004) on a global scale.*

3. *In order to address trade-offs between human and ecological water demands, multiple stressors on human water security and ecosystem conservation need to be considered. The applied approach is able to consider different drivers of change such as dam operation, water use and climate change.*

4. *Next to the flow regime modifications, the threat for riparian wetlands also depends on the society's capacity to act to the changes. This kind of threat has not yet been taken into account in large scale studies. In order to fill this gap, we combined quantitative with qualitative results. The implementation of counteractive measures depends especially on the legal and institutional framework in place. Therefore, we collected 6 different criteria (legal environmental flow provisions, presence of RBOs, at least one relevant treaty, and specific treaty provisions such as water allocation mechanism, conflict resolution mechanism, and flow variability management). In addition, new dam construction is likely to further modify flow regimes in the future, but currently no large-scale dataset on major dam initiatives (including planned storage capacities) is publicly available. Therefore, we collected the number of dams that are currently planned, proposed or under construction in the upstream areas to give a first indication, where future dam construction is likely to affect the inundation of specific riparian wetlands.*

5.   *Our discharge simulations were done on a daily time-step. This is important as many ecological functions and habitats are facilitated by hydrological events that last only up to some days (e.g. strong precipitation events, bankfull flow, and flood formation).*

6.   *Today, river flows are considerably affected by human activities worldwide, and the speed of river ecosystem destruction and biodiversity loss is exceeding the ability of scientists to review applied water management practices and ecological consequences for each river. Therefore this study assesses flow regime modifications on a global scale. The approach is performed on a detailed river network with a spatial resolution of 5x5 arc minutes and can be applied for single reaches of larger rivers with a global coverage.*

7.   *The approach will allow new applications related to riparian wetland flooding. Examples include the quantification of specific ecosystem services provided by intact riparian wetlands (e.g. forest production, water purification, fish production, flood control, etc.) and how this is likely to change in the future. The framework could support policy makers at international level (e.g. at forums like UNEP, OECD, European Union, Convention on Wetlands of International Importance, and Convention on Biological Diversity) in balancing water allocations to humans and nature, implementing global conservation efforts, and planning of water infrastructure location and design.*

2. It is stated in the introduction (p. 2, line 26) that a new approach is needed to water resources management, which among other things should allow for sufficiently high flows for sustaining floodplain wetlands. However, in line with comments of bullet point 1 (above), this proposed novelty remains unclear to the reader. For example, haven't we gained some relevant knowledge from regulation schemes applied to the principal Colorado River in the US (Stevens et al., 2001; Stromberg et al., 2007; Cross et al., 2011)? These schemes have included controlled floods as part of the strategy to minimise adverse impacts to downstream ecosystems. Perhaps there other relevant examples.

*Thanks for this remark. The aim of this paragraph was not to claim that the "new approach" on water resource management is our idea. Rather we wanted to state that both flood protection for people and controlled floods for riparian wetlands are important and need to be considered in practice within the framework of integrated water resource management. We appreciate the given examples from the literature and also wanted to include them in the paper as good examples for eFlow (controlled flood) provisions in practice. However, in the context of shortening the introduction, we totally removed these sentences.*

3. The language of the manuscript is overall good. There are some exceptions though, including the introduction. In particular, the research questions and the related text include awkward formulations (e.g., multiple sentences starting with Thereby. . ./ Therefore. . .), please check.

*We agree that the language of this manuscript can be improved. We checked again grammar, formulations and word spelling. Thanks for the given examples, which were corrected. The research questions were revised as well.*

4. There is a lack of results comparison to previous studies in the discussion section, which should be addressed before publication. The now included references do mainly not relate to the results (study outcomes) and need therefore to be complemented. For instance, are the results regarding impacts on the 93 Ramsar wetlands in different world regions (p. 17, lines 3-11) consistent with previously

reported results for these regions? Alternatively, do the results partly contradict or point to new and previously unnoticed aspects? (Also, the reader is not well informed about the existence or absence of similar studies, see bullet point 2 above regarding the introduction). The same questions can be asked for other key results, such as impacts of climate change and the related identified hotspots (p. 17, line 12-15), and competition of water (p. 17, lines 31-32). Overall, the discussion section is rather general and would benefit from an extended discussion of results. The aims of the study need not to be reiterated in the beginning of the discussion section.

*In order to address this comment, we included more comparisons of our key findings in the results chapter with existing regional (Georgiyevsky et al., 1995; 1996; 1997; Kingsford, 2000; Khublaryan, 2000, Uluocha and Okeke, 2004; UNEP, 2008; Middelkoop et al., 2015) and large-scale (Grill et al. 2015; Schneider et al., 2011b; 2013; Tockner and Stanford, 2002; Dankers et al., 2013, Zarfl et al., 2014) studies. We also formed a bracket by addressing some of the studies already mentioned in the introduction section.*

*Under 'Discussion', we explicitly discuss now the limitations of our quantitative-qualitative approach and the consequences of our decisions made under the methodology chapter. Afterwards we describe what the results mean for water management depending on the nature of the threat. Aims and main results of the study as well as general descriptions were removed in the discussion section.*

5. The main conclusions of the paper are not clearly presented. Maybe a separate conclusion section could help?

*Thanks for this remark. In our revised manuscript we put particular attention on the revision of the discussion and conclusions section. We more clearly present now the novelty and limitations of our study. As suggested, we created a separate conclusion section. Here, we included a sub-section on future research and the potential of our approach to be applied to similar research questions related to riparian wetlands. Unfocused conclusions were removed.*

**Response to Referee #3**

*Dear Referee #3,*
*we are very thankful for your profound evaluation of the paper and helpful comments, which further*
*improved this paper. Please find below our point-by-point responses describing our revisions*
*(highlighted in blue and italic type).*

The authors aim to address an important issue: Identifying Ramsar riparian wetlands that exhibit current and future variations in ecologically consequential inundation patterns as a result of human-modified flows (e.g., dams). They ask three particular research questions to best identify these wetlands. These questions focus on the impact of current water resource management on riparian wetland flows, the effect of future climate change on inundation of these wetlands, and the implications of low government and societal infrastructure and capacity to make changes to future management.

The goal and research questions the authors attempt to address are broad and could be impactful if addressed and translated well. However, a major revision is required to ensure both the quantitative work behind the research and the communication of this work is effective. Below, I provide major suggestions for the manuscript followed by some general comments.

Major Point 1: The Introduction reads somewhat like a full literature review that continues for quite some time without a direct point. It was well into the sixth paragraph that the goal and research questions were stated. I would suggest tightening up the Introduction, providing only key points throughout, and early on (perhaps at the end of the first paragraph) allude to the main point of the paper (e.g., "We aim to. . ."). Then, the authors can safely state the full objective and research questions at the end of the Introduction.

*We agree that the introduction was too long and more like a literature review. Accordingly we*
*condensed the text by often providing only key points and referring to the literature. The detailed*
*descriptions of the flood pulse concept and the paragraph on ecosystem services of wetlands were*
*removed. However, following the description of the current situation and providing the rationale for*
*our indicators, we added an additional paragraph to the introduction. This paragraph mentions*
*valuable studies of the past addressing flow regime alterations on larger-scales and highlights the*
*novelty of our study.*

Major Point 2: Something is very misleading and incorrect about discussing a "natural" flow regime in knowingly modified watersheds and aquatic systems. Also, the word "natural" is used throughout the Abstract and Introduction (ala Poff et al. 1997), and it is not until the Methodology that authors define natural flow. The authors describe natural flow for this paper as "simulated taking into account current climate and landcover conditions, but no further anthropogenic impacts." This, by no means, would constitute a "natural" flow regime as described in past literature. I would recommend modifying terminology and the discussion throughout the paper to consider this as your "baseline" flow regime from which the analyses aims to understand current water resource management implications on the riparian wetlands and project changes of these regimes due to climate change

*We agree that the use of the terminology 'natural flow regime' is misleading due to the fact current*
*climate and land cover conditions are contained. Therefore, we make use of the terminology*

*"reference flow" now. The new terminology is applied throughout the manuscript. Under Chapter 2.1.3 we explain the meaning of the reference flow which shall reflect near-natural flow conditions where anthropogenic impacts such as dam operation and water use were not considered. But current climate and land-use are taken into account.*

Major Point 3: The goals of the paper and research questions are poorly worded need more information. What, specifically, are the "riparian wetlands?" In the Abstract, the authors suggest they look at 93 Ramsar sites. Are the "riparian wetlands" the "93 Ramsar riparian wetlands?" For Research Question 1, why are 6025 dams selected? Are these dams specifically located upstream of Ramsar riparian wetlands? What are the "different water use sectors"? Also, delete "Thereby" at the beginning of the second sentence. For Research Question 2: "Inundation" cannot be "impaired" because "inundation" does not necessarily denote a positive quality. The authors could replace "impaired" with "exacerbated or diminished" or "modified." Also, delete "Therefore" at the beginning of the second sentence. Research Question 3 is stated in a grammatically incorrect way, so it took a few re-reads to understand it. Move "could" after the word "sites." Also, what is a "low capacity to act?" This is definitely not clear.

*Thanks for these valuable corrections. We made the following changes in the text:*

- *Due to comments from Reviewer #1, we put the screening tool more in the focus of this paper (rather than the analysis). In this context, we explain now and make clear from the beginning of this manuscript that the proposed approach is exemplified on 93 riparian wetlands of international importance (Ramsar sites), but can be applied on global scale taking into account all larger riparian wetlands. We also make clearer now that the 6025 dams are allocated to the global WaterGAP3 stream net (Chapter 2.1.1).*
- *In order to assess flow alterations due to dam operation, the number of dams implemented in the model has been further increased. We operate now >6000 dams which is state-of-the-art in comparison to other global models (see Haddeland et al., 2013, supporting information). In order to have a clear cut, we decided to consider all large dams + smaller dams with a storage volume >0.5 km$^3$.*
- *The different water use sectors are described in Chapter 2.1.1*
- *The proposed corrections for grammatically incorrect sentences and incorrect word use were adopted. Thanks for the examples.*

Major Point 4, WaterGAP3 runs: Streamflow, for what the authors term "daily natural flow regimes" (1981-2010), is simulated with 2004 land cover. Using 2004 land cover is okay; however, going back to the use of the word "natural". . .how can this be considered natural flow when the landscape for each area is likely highly modified and streamflow is a reflection of these anthropogenic activities? Also, there is no mention of calibration and verification of the model, which admittedly would be difficult a global scale. Therefore, is the entire paper a thought experiment using an uncalibrated global model to help explore hypotheses? It would be okay if so, though this framework should be characterized as such throughout the paper. Also, the results (maps, in particular) should emphasize the paper's overarching approach (i.e., the thought experiment – a "screening tool" is mentioned in the Discussion, hypotheses testing, and/or a conceptual model). If calibration and verification did occur at some stage and is not referenced, again, measured streamflow would reflect the managed conditions, not some unattainable "natural" or "near natural" condition. The model scenarios are

therefore a bit confusing and need some rethinking, definitely in the presentation of what they are but potentially in which ones should be used. For example, consideration should focus on whether only the managed scenario and future climate/management conditions should be used since the true "natural flow regimes" aren't captured.

Also, the authors talk about the database of dams that are used, but how does that relate back to the Ramsar wetlands? Are these dams all upstream of Ramsar wetlands? As I read on, it became a bit clearer that this is simply a global database, and Ramsar wetland areas within the global domain are analyzed. However, this information (spatial domain and selection of dams) needs to be clearer up front.

The authors likely have all the information mentioned in this Major Point. There is simply a need for better and clearer communication regarding these bits of information. As a result, the Methodology section seems quite disjointed and leaves the reader guessing at how the authors conducted the analyses.

*Thank you for these valuable comments. The WaterGAP3 model is calibrated and validated which was described in the manuscript at p.7 line 32 to p.8 line 3. For the verification of the model, we refer to p.8 line 15 and the reference Schneider et al. (2011a) given in the text, which contains details of the model performance with regards to bankfull flow events. However, given this comment we totally understood that the structure of the Methods section and the model description needed to be improved. We addressed this comment by including sub-headings, a flow chart schematically illustrating the single steps of our approach, and an extra paragraph on model calibration and validation in order to increase clarity, transparency and understanding of the paper. Furthermore we added two more references on WaterGAP3's ability to represent maximum flow magnitudes (see validation in Schneider 2015, Eisner 2016). The entire paper is far from being a thought experiment, although any model experiment could be understood as a 'thought experiment'. Hydrological models are useful tools to mirror the reality, i.e. river discharge, in an abstract manner. The higher the agreement of simulated and observed data records the better the model performance. In the calibration process, WaterGAP3 model simulations take into consideration human impacts in terms of managed reservoirs and dams, water abstractions and return flows from 5 different sectors, urban water transfers and land use conditions. The model is calibrated against an observed discharge record by adjusting one free parameter (runoff coefficient) and validated to an independent period of the same discharge record. Based on the calibrated model (i.e. with adjusted runoff coefficient) the 'natural flow' is represented by a model simulation driven solely by the meteorological forcing of the respective time period, i.e. 1981 to 2010. Human interventions in form of managed dams and reservoirs, water abstraction, return flows, urban water transfers are omitted in this model simulation. This approach as well as the terminology 'natural flow' is commonly used in the community of global hydrological modellers. However, recognising the misunderstanding of the information given in the text we improved the manuscript by providing model-specific information and references to the model calibration and validation, and conducted a thorough revision of the entire text. Furthermore, we changed the terminology from 'natural flow' to 'reference flow' (see 'Major Point 2').*

*The dam database (GRANd) used by the model is initially independent of the Ramsar wetlands. GRANd contains the information on the location, storage capacity and main purpose of the largest dams of the world which are not necessarily located upstream of Ramsar wetlands. We decided to*

*focus on the Ramsar wetlands to exemplify our screening tool and because of their importance and description found in scientific literature. We now communicate this in a clearer way in the text.*

Major Point 5, Discussion and Conclusions: Be careful here. Because this is thought experiment using a global model (again, unless calibration/verification happened but wasn't mentioned), your conclusions need to be balanced with a statement of the conceptual aims of the paper and associated limitations/assumptions. The quantitative analyses isn't incredibly quantitative, and I wince a bit with the use of numbers like "8% are significantly impaired" and flood volume is likely to be decreased at 41% of the sites. . ." when those are all relative numbers with no basis in reality. Please mention up front in the conclusions or make a separate section of the limitations and assumptions with regard to what the analyses can actually provide.

*Thanks for this remark. We revised the discussion section by explicitly discussing the limitations of our approach and the consequences of our assumptions made. This is followed by paragraphs describing what the results mean for water management depending on the nature of the threat. The calibration and validation is described in an extra paragraph now.*

Major Point 6: In general, the English is okay as written. However, it's important that someone extremely proficient in English re-review this paper for odd placement of verbs, adjectives, modifiers, etc., and poor word selection. One small example, on Page 5, Line 14 "For Europe, a higher number of sites "were gained" as the European wetland geodatabase. . .". This should be "were selected" or "were chosen". There are many instances like this throughout the paper, and I do not list them all below.

*We checked again grammar, wording and spelling. Thanks for the given examples.*

Specific Comments

Page 1, Line 9 – Recommend changing all references of "mankind" to "humankind" and "man-made" to "constructed"

*Thanks, we modified all terms, respectively.*

Page 1, Line 9 – These eco services are provided not only via the regular patterns of inundation but also regular patterns of drying – so actually, it's the \*variability\* inundation patterns that is important.

*Thanks, we included 'regular patterns of inundation and drying' to be more precise.*

Page 1, Line 26 – Need to review and add Dixon et al (2016) as well. Dixon, MJR, Loh J, Davidson NC et al. 2016. Tracking global change in ecosystem area: The Wetland Extent Trends index. Biol. Conserv. 193: 27-35.

*Thanks, we included this reference.*

Page 2, Lines 18-19 – Is this true for all "larger cities?" What spatial scale is this referring to? Are these global or regional estimates? If regional, what regions?

*Thanks for this comment. In the context of shortening the introduction, we needed to delete the part on city water transfers.*

Page 2, Line 25 – Again, what are "natural sites?"

*Thanks, we exchanged 'natural sites' with 'pristine and not heavily altered floodplains'.*

Page 2, Line 30 – Not all floodplains are wetlands, which is how this sentence reads. Please correct.

*Thanks, the sentence was corrected.*

Page 2, Line 32 – What ecological processes are initiated? Some of these processes may be initiated by drying not wetting.

*Thanks for this remark. In the context of shortening the introduction, we needed to delete detailed descriptions of the flood pulse concept. However, we made clear in the text from the beginning (abstract) that periodic flooding and drying is responsible for the generation of floodplain ecosystems.*

Page 3, Line 2 – What is engendering what? This clause doesn't make sense.

*Thanks. This sentence was rephrased to make clear that "the periodic occurrence of overbank flows engenders one of the most dynamic, diverse and productive systems in the world".*

Page 3, Line 4 – That's a very broad statement, that all floodplain wetlands contain more species than any other landscape unit. Need more specifics here because it's likely not what the authors intended to say.

*Thanks. In the context of shortening the introduction, we needed to delete this part on ecosystem services.*

Page 3, first paragraph – The Roman numerals are not needed when providing full sentences after them. Suggest removing all Roman numerals here.

*Thanks, we agree on removing the Roman numerals. However in the context of shortening the introduction, we needed to delete this part.*

Page 3, Line 24 - What are "fellow riparians?" Please be more specific.

*Thanks, we replaced it by "upstream/downstream water users"*

Page 3, Line 25 – What projections? Please be more specific.

*Thanks, we rephrased the sentence to be more precise.*

Page 4, Lines 12-16 – Break up this sentence into two or more sentences.

*Thanks, we broke up the sentence.*

Page 5, Lines 19-20 – These sentences can be deleted and are unnecessary.

*Thanks, sentences were deleted.*

Page 5, Line 23 – "percent change in flood volume": from what period to what period? Please provide time frame.

*Thanks for this remark. We included now a flow chart under methodology which clearly illustrates the time periods considered for each model run. The time frames are also provided under each sub-heading.*

Page 5, Line 28 – It is not clear at this point what "sufficient capacity to act" means. Suggest modifying this or adding some clarification here to lead the reader to the more specific methods discussion.

*Thanks, we rephrased the sentence.*

Page 5, Line 3 – The simulation of daily natural flow regimes would still be an expression of a modified landscape, so how are these natural?

*See Major point 3. We are using the term 'reference flow' now.*

Page 5, Lin 9 – Need clarification of what type of "daily river discharge" is being simulated here – "natural" or "managed"? (After reading on, it becomes obvious it's "natural" but that needs to be mentioned straight away.)

*Thanks. We included subheadings and a flow chart in order to better structure the methodology section and be more clear which model run is described.*

Page 5, Line 20 – Switched to "near-natural" from "natural" in this sentence. Please be consistent.

*Thanks, we are using the term 'reference flow' now throughout the paper.*

Page 7, Lines 6-9 – Need to be clear here why the simulation includes these specific 6025 dams. Why were they chosen? Intuition would tell me they are all upstream of Ramsar sites, but further reading seems to suggest that they are simply part of the global database. These questions regarding methods also suggest that clear summary statements of what the quantitative analyses is up front in the Methodology should be added – meaning state your steps: exact simulations, the spatial scale, how dams were selected, how the Ramsar sites were overlain on the global map, etc. Then, details can be added after this summary.

*Thanks. Under Chapter 2.1 we describe now up front WaterGAP3 in general and provide a flow chart schematically illustrating the modelling exercises. Under the following subheadings, specific details for each model run are described. All subheadings are in compliance with the items in the flow chart. Additionally, we provide more rationales under methodologies for our decisions (dam selection, scenario selection, model selection, ...). The 6025 dams are taken from the GRANd database, which were allocated to the global grid cell raster of WaterGAP. The proposed screening tool can be applied on global scale and is only exemplified on the 93 Ramsar sites. We explain this more clearly now in Chapter 2.1.1.*

Page 9, line 14 – What selected sites? The Ramsar wetlands? Again, details are needed here.

*See Major Point 3.*

Page 9, Lines 14-15 – This sentence is a bit wonky and needs to be reworded.

*Thanks, we reworded the sentence.*

Page 9, Line 18 – How were the cutoff thresholds for Table 2 selected?

*Thanks, we included the missing information.*

Page 9, Line 20 – Again, clarify what the "low capacity to act" is.

*Thanks. We explain 'capacity to act' now in the Chapters introduction and methodology. Under 2.2.2 we describe how we defined the cutoff thresholds.*

Page 9, Line 27 – Define blue water

*Thanks for this remark. The term "blue water footprint" is used in the cited literature and probably unnecessary jargon. Hence, we rephrased this and describe now that the scarcity threshold is reached when 20% of the streamflow is depleted.*

Page 9, Line 30 – Again, how were the Table 3 thresholds derived?

*Thanks, we include the missing information.*

Page 10, Lines 25-27 - Cut these sentences. Too much introduction here.

*Thanks, we agree and removed the mentioned sentences.*

Page 11, Line 4 – These wetlands are "moderately impacted" – as far as the map seems to read.

*Thanks. We are more specific now about the locations. Here, Southern Europe was exchanged by Iberian Peninsula.*

Page 11, Line 17 – N=2, though, correct? So this is only discussing two wetlands, right?

*Thanks, we deleted that statement.*

Page 12, Line 28 – Is this the ensemble median for the GCMs as input to the WaterGAP3 model or the ensemble average of the output of the WaterGAP3 model?

*Thanks for this remark. We rephrased the text accordingly. It's the ensemble average of the output of the WaterGAP3 model. We included 'ensemble median derived from five hydrological simulations driven with different GCM-projections'.*

Page 13, Liens 16-17 – Now that is a very interesting finding!

*Thanks.*

Table 1, change "not/slightly" to "none/slightly" – same with the figures: "not/slightly" does not make sense.

*Thanks, we changed the text as suggested.*

Table 2, delete "the number of" in the caption.

*Thanks, we deleted it.*

Table 4, define "formal institutional capacity" in the caption to make the table stand alone.

*Thanks, we included the missing information to make the table stand alone.*

The final edits for the paper are included in the Major Points listed previously.

[revised manuscript text omitted]

Tier 1: Simulation of flow regimes modified by current water management (2000s)

Tier 2: Simulation of flow regimes modified by future climate change (2050s, w/o water management)

Tier 3: Simulation of reference flow regimes (2000s, near-natural)

Tier 4: Estimation of bankfull flow (=starting point of inundation)

Tier 5: Assessment of overbank flows (change in flood volume between modified and reference flows)

**Figure 1:** Schematic illustration of

[Figure]

[Figure]

**Figure 2: Global map of overbank flow alterations for selected riparian wetlands of international importance (#1-93) as a consequence of current water resource management.**

[Figure]

**Figure 3: Global map of overbank flow alterations for selected riparian wetlands of international importance (#1-93) as a consequence of** the exclusive effect of climate change in the 2050s.

[Figure]

**Figure 4: Potential impact** of new dam initiatives taking into account  dams currently planned or under construction in the upstream area of each riparian wetland.

[Figure]

**Figure 5: Current capacity to act in regard to anthropogenic flow regime modifications for selected riparian wetlands. The left semicircle represents the water availability for ecological allocations, while the right semicircle characterizes the institutional capacity in the upstream area. For wetlands with a non-transboundary upstream area (white border), the right semicircle represents  presence or absence of legal provisions or official recommendation to establish eFlows.**

---

## Referee Report (RR1)

[revised manuscript text omitted]

1) This table does not match the text. The text says no dams = low impact. Here no dams = zero impact.
2) The text says the latter 2 categories have roughly the same # of wetlands. I would like to see the actual # of wetlands in each category either in the text or in the table.
3) I actually don't see it as necessary to begin with, the text does a fantastic job of explaining your classification.

**Table 3 : Water availability for ecological allocations defined by means of the number of months with water scarcity upstream of the Ramsar site.**

| Number of months with water scarcity | Water availability for ecological allocation |
| --- | --- |
| 6 – 12 | LOW |
| 2 – 5 | MED |
| 0 – 1 | HIGH |

**Table 4 : Institutional capacity in place in transboundary upstream areas of riparian wetlands based on formal arrangements such as international water treaties, river basin organizations, legal eFlow provisions and specific treaty provisions.**

| Score | Institutional capacity |
|-------|------------------------|
| 0 – 2 | LOW |
| >2 – 4 | MED |
| >4 – 6 | HIGH |

[Figure]

**Figure 1: Schematic illustration of the steps taken in the quantitative analysis based on WaterGAP3 modelling.**

[Figure]

I'm thrilled to see the flow chart. Excellent work.

**Figure 2: Global map of overbank flow alterations for selected riparian wetlands of international importance (#1-93) as a consequence of current water resource management.**

[Figure]

**Figure 3: Global map of overbank flow alterations for selected riparian wetlands of international importance (#1-93) as a consequence of the exclusive effect of climate change in the 2050s.**

[Figure]

**Figure 4: Potential impact of new dam initiatives taking into account dams currently planned or under construction in the upstream area of each riparian wetland.**

[Figure]

**Figure 5: Current capacity to act in regard to anthropogenic flow regime modifications for selected riparian wetlands. The left semicircle represents the water availability for ecological allocations, while the right semicircle characterizes the institutional capacity in the upstream area. For wetlands with a non-transboundary upstream area (white border), the right semicircle represents presence or absence of legal provisions or official recommendation to establish eFlows.**

---

## Author Response (AR2)

**Response to Report #1**

*We are very thankful for the positive feedback of Reviewer #1 and the profound and helpful evaluation of the paper during the review process, which further improved this paper. Please find below our point-by-point responses describing our revisions in detail (highlighted in blue and italic type).*

And I mean extremely minor revisions. I caught a few typos/misused words.

I wish there was a way to include the large Table 2 & 3 in the main body of the paper. These tables would be a huge 'hook' to interest casual readers by showing them what is going on with the Ramsar wetland in their 'back yard'. They are very large though. So I suppose it is what it is.

My one substantive regret is that the researchers never compared their simulated overbank flows to observed conditions the past decade or so. That would have left no doubt in my mind their modeling was solid. As it is, I have to trust the model. This is not a show-stopper, just a gripe.

*Simulated overbank flows were compared to observed conditions. We refer to the related reference (Schneider et al., 2011a) in the text, which describes in detail the approach including a validation of bankfull flow estimates against observed conditions. In response to this comment, we want to be more clear on this and added "against observed conditions" to the related text in the manuscript as well as specific validation results of the related reference.*

Aside from this one regret, the revised paper is fantastic. The authors addressed all my concerns and did it in an elegant fashion. I give this paper two enthusiastic thumbs up. I left many comments embedded in the attached PDF of the revision.

*We are very thankful for the additional comments of reviewer #1 in the attached PDF. We addressed all comments in the revisions of the manuscript and particularly made the following changes:*

- *The introduction has been further shortened a bit by removing dated references/ statements in a paragraph on page 3.*
- *We divided longer paragraphs into two paragraphs as suggested.*
- *We provide reasons for the cutoff threshold defined in section 2.2.1. In addition, we included the number of affected wetlands for each class in Table 2.*
- *All comments addressing sentence and paragraph structure, spelling and wording were adopted as suggested.*

**Response to Report #2**

The introduction has now been carefully revised to explain relevant state-of-the art knowledge and knowledge gaps, motivating the objectives of the study.

Regarding the lack of results comparison to previous studies, the issue was addressed by adding relevant references and comparing the present outcome with these references. It was not consistently done by using the discussion-section as suggested – sometimes comments were made in the results section too. However, this is more a matter of style, which is up to the authors (as long as the clarity is not affected). Other relevant changes were made in the discussion section, perhaps in response to other review questions. The end result was in any case a needed improvement: looks clear to me.

The authors addressed also the previously unclear conclusion section, which now relates to the presented results in an adequate way.

This means I would recommend publication of the manuscript in its present form.

*We are very thankful for the profound evaluation of the paper and helpful comments of Referee #3 during the review process, which further improved this paper.*

**Response to Report #3**

*We are very thankful for the profound evaluation of the paper and the helpful comments of Referee #2, which further improved this paper. Please find below our point-by-point responses describing our revisions (highlighted in blue and italic type).*

I previously reviewed this manuscript and would like to compliment the authors on a much-improved version. The clarity of the revised methods greatly assists in fully understanding their objectives and approaches, thereby increasing the potential impact of the paper. The authors' synthesis of worldwide patterns of current and future threats to Rasmar wetlands as a result of human-modified flow regimes (e.g., dams) and use of a global hydrological modeling framework to do this should result in a highly-cited paper.

Below, I provide minor suggestions for edits to the manuscript:

General comment: The authors should discuss how the availability of data on different continents and in different countries might have affected their model simulations and findings. One might expect data availability is imbalanced among, for example, wealthier and poorer countries. This hits upon the topic of how uncertainty is addressed in the modeling effort, which is addressed in some capacity in the Discussion. Because the change in hydrology are presented as relatively changes and in "bins" of magnitudes of change, output uncertainty may not be as important as the input uncertainty – such as that of data availability. If there is a clear mention that this particular uncertainty exists, it should be sufficient for the purposes of a global scale analyses.

*Thanks for this remark. We clearly mention now the existence of this particular uncertainty by the inclusion of the following text in the discussion section: "We stress that global scale modeling is limited by the quality of all input data used in our calculations. In wealthier countries data availability and verification is often more advanced. Hence, a bias of the model performance can be expected with a higher uncertainty in data poor regions. However, our global scale modeling approach allows transferring knowledge to these regions by identifying hotspots of risk where further hydro-ecological research can be directed."*

Page 1, Lines 26-27. Revised the first sentence of the Introduction. It is not grammatically correct. What about, "Natural wetland areas have declined at the global scale by 31% between 1970 and 2008 (Dixon et al. 2016), and even higher numbers are likely for floodplain wetlands specifically".

*Thanks, the sentence was changed as suggested.*

Page 3, Lines 10-30. This paragraph seems really long. Suggest making a new paragraph starting on Line 15 to introduce the study's goals.

*Thanks, the paragraph was divided as suggested.*

Page 3, Line 25: Fully define what you mean by "society's capacity to act" here. It's too ambiguous without a definition.

*Thanks for this comment. We included an additional sentence to be more precise on what we mean by "society's capacity to act" here.*

Page 4, Line 33: What is the parameter that's adjusted in the model. It's pretty amazing to have only one parameter to adjust.

*Thanks, we included the missing information. The parameter is runoff coefficient γ.*

Page 4, Line 39: Provide (parenthetically) the anthropogenic impacts that are removed from the model for these runs. They can be explained further in subsequent sections, but it's helpful here to briefly mention them.

*Thanks, we added the information.*

Page 4, Line 27: Change "considers now" to "now considers".

*Thanks, we corrected the sentence as suggested (Page 5, line 27).*

Page 8, Line 29: Define RBO here rather than on Line 33.

*Thanks, we now define RBO at the first occurrence (i.e. Page 8, Line 29).*

Page 11, Line 25: Change "on global scale" to "at global scales".

*Thanks, the sentence was corrected as suggested.*

Page 12, Line 26: Restate what the two sites are here.

*Thanks, we restate the two sites now.*

Page 12, Line 35: Start a new paragraph here. This paragraph seems too long.

*Thanks, the paragraph was divided as suggested.*

Page 13, Lines 5-15: This seems to be repetitive and was, in part, discussed on Page 12. Please revise, potentially by merging the two discussions on capacity to act and counteractive measures.

*Thanks, we removed repetitive sentences at page 13.*

Page 13, Lines 5-35: There are a lot of grammatical errors and odd sentence structures in this section. Please reread and revise.

*Thanks for this remark. We did our best as non-native English speakers to revise this section.*

[revised manuscript text omitted]